# Characterization of convergent thickening, a major convergence force producing morphogenic movement in amphibians

**David R Shook[1,2]\*, Jason WH Wen[3], Ana Rolo[4], Michael O'Hanlon[2], Brian Francica[5], Destiny Dobbins[6], Paul Skoglund[1], Douglas W DeSimone[2†], Rudolf Winklbauer[3†], Ray E Keller[1,2†]**

[1]Department of Biology, University of Virginia, Charlottesville, United States; [2]Department of Cell Biology, University of Virginia, School of Medicine, Charlottesville, United States; [3]Department of Cell and Systems Biology, University of Toronto, Toronto, Canada; [4]Centre for Craniofacial and Regenerative Biology, King's College London, London, United Kingdom; [5]Aduro Biotech, Berkeley, United States; [6]Independent researcher, Philadelphia, United States

**\*For correspondence:**
drs6j@virginia.edu

†These authors are co-senior authors to this work

**Competing interest:** The authors declare that no competing interests exist.

**Summary** The morphogenic process of convergent thickening (CT) was originally described as the mediolateral convergence and radial thickening of the explanted ventral involuting marginal zone (IMZ) of *Xenopus* gastrulae (Keller and Danilchik, 1988). Here, we show that CT is expressed in *all* sectors of the *pre-involution* IMZ, which transitions to expressing convergent extension (CE) *after* involution. CT occurs without CE and drives symmetric blastopore closure in ventralized embryos. Assays of tissue affinity and tissue surface tension measurements suggest CT is driven by increased interfacial tension between the deep IMZ and the overlying epithelium. The resulting minimization of deep IMZ surface area drives a tendency to shorten the mediolateral (circumblastoporal) aspect of the IMZ, thereby generating tensile force contributing to blastopore closure (Shook et al., 2018). These results establish CT as an independent force-generating process of evolutionary significance and provide the first clear example of an oriented, tensile force generated by an isotropic, Holtfreterian/Steinbergian tissue affinity change.

## Editor's evaluation

In this manuscript Shook and collaborators investigate the process of convergent thickening, a morphogenetic process occuring within the mesoderm prior to involution during amphibian gastrulation. They demonstrate that convergent thickening occurs in the involuting marginal zone, and acts as an isotropic, symmetrically acting morphogenic process that can close the blastopore. They demonstrate that convergent thickening proceeds before and is independent of convergent extension and is also independent of dorsoventral patterning. Whereas the cellular and molecular basis of convergent thickening remains to be discovered, this process is driven by increased interfacial tension between the deep and superficial epithelial layers of the involuting marginal zone.

## Introduction

Morphogenesis is driven by force-generating, local cell behaviors, integrated to form that shape tissues. We understand much about the regulation of embryonic patterning and have made progress

in understanding how patterning regulates specific morphogenic machines. But understanding the Mechanome (*Lang, 2008*), the molecular and cellular mechanisms of these machines, how they generate tissue level mechanical properties and forces, and how these forces interact in the global biomechanical context to drive embryonic shape changes, is the challenge ahead. We explore these issues in gastrulation of an amphibian, *Xenopus laevis*.

Gastrulation involves movement of the cells of presumptive internal tissues from the outside surface through the blastopore to the inside of the embryo, thereby establishing the fundamental structure of the three-layered adult body plan. One category of movements involves the 'convergence' or decrease in circumference of the involuting marginal zone (IMZ), a ring of tissue around the vegetal endoderm (VE) (*Figure 1A*, St 10-; *Video 1*\*, top). The cells within the IMZ actively converge to generate a constricting force around the VE, which rolls the IMZ forward and inside (Involution; *Figure 1A, B*, red arrows) (*Keller et al., 2000*; *Keller et al., 2008*; *Keller and Sutherland, 2020*). After involuting, the early-involuting tissues originating from the lower (vegetal) region of the IMZ migrate across the blastocoel roof (*Figure 1A*, St 10 + and 11–11.5; brown cells, blue arrows) whereas the later-involuting tissues, the presumptive somitic and notochordal mesoderm, originating from the upper (animal) region of the IMZ (*Figure 1A*, purple and red cells) show narrowing (convergence) in the mediolateral axis and a commensurate lengthening (extension) in the orthogonal anterior-posterior axis (CE)(*Figure 1A*, St 11–11.5, green and light blue arrows). CE of these tissues, along with a roughly congruent CE of the overlying posterior neural plate (dark blue cells, *Figure 1A*, St 11–11.5), which lies within the ring of tissue above the IMZ, the non-involuting marginal zone (NIMZ), elongate the body axis.

The forces driving convergence of the IMZ as it closes the blastopore and moves over and internalizes the VE are generated by two morphogenic machines in the IMZ, CE and CT. The mechanism and function of CE have been characterized (Reviewed in *Keller et al., 2000*; *Keller et al., 2008*; *Keller and Sutherland, 2020*) but little is known about CT, and the relationship between CT and CE remains unclear. CE begins at mid-gastrulation and is driven by an oriented, force-producing cell motility, *mediolateral intercalation behavior* (MIB; *Figure 1C*; *Wilson and Keller, 1991*; *Shih and Keller, 1992a*; *Shih and Keller, 1992b*; *Pfister et al., 2016*) within the presumptive notochordal and somitic mesoderm. In addition to displaying CE in vivo (*Vogt, 1929*; *Keller, 1976*), movements of these tissues in explants of the dorsal marginal zone (DMZ, = IMZ + NIMZ) mimic those in vivo (*Schechtman, 1942*; *Keller and Danilchik, 1988*; *Figure 1B*, Dorsal quadrant, blue and green arrows).

In contrast, CT was initially described as a tendency for explants of the ventral sector of the early gastrula to converge and thicken and eventually form a mini-gastroid lacking both dorsal tissues and CE (*Figure 1B*, Ventral Quadrant, white arrows) (see *Keller and Danilchik, 1988*). This led to the incorrect assumption that CE was a property of the dorsal sector of the IMZ and that CT was a property of the ventral IMZ (*Keller and Danilchik, 1988*).

However, several observations suggest that CT is an independent morphogenic process that begins before CE and is initially expressed throughout the IMZ. First, *Scharf and Gerhart, 1980* showed that *Xenopus* embryos ventralized by UV irradiation of the vegetal side before first cleavage closed their blastopores, albeit symmetrically, despite lacking dorsal tissues expressing CE. This suggested that the 'CT' expressed in ventral explants of the normal gastrula (*Keller and Danilchik, 1988*) might be an independent morphogenic process acting alone to close the blastopores of ventralized embryos. Second, ventralization rescued blastopore closure in myosin heavy chain IIB morphant embryos, suggesting that the mechanism of this symmetrical blastopore closure in ventralized embryos differs from that driving CE (*Rolo, 2007*). Third, recent work (*Shook et al., 2018*) confirmed that ventralized embryos close their blastopores isotropically (see *Video 1*\*, bottom), showed that 'giant' sandwich explants from ventralized embryos express CT throughout the IMZ (*Figure 1D and E*; white arrows), and showed that they generate 2 µN of convergence force during gastrulation. Moreover, giant sandwich explants made early from normal embryos (e.g. *Figure 1C*) generate tensile convergence forces of 0.3 µN or more during the two hours *prior to the onset* of CE (*Shook et al., 2018*), at Stage 10.5 (*Shih and Keller, 1992b*; *Lane and Keller, 1997*). Together, these results demonstrate that CT is a force generating process independent of CE and suggest that it is initially expressed by the entire IMZ. Lastly, convergence around the blastopore or within giant sandwich explants prior to the onset of CE movements in other amphibians (Shook, unpublished observations) and delayed expression of CE until near the end of blastopore closure in some anurans (*Benítez and Del Pino, 2002*; *Del Pino et al.,*

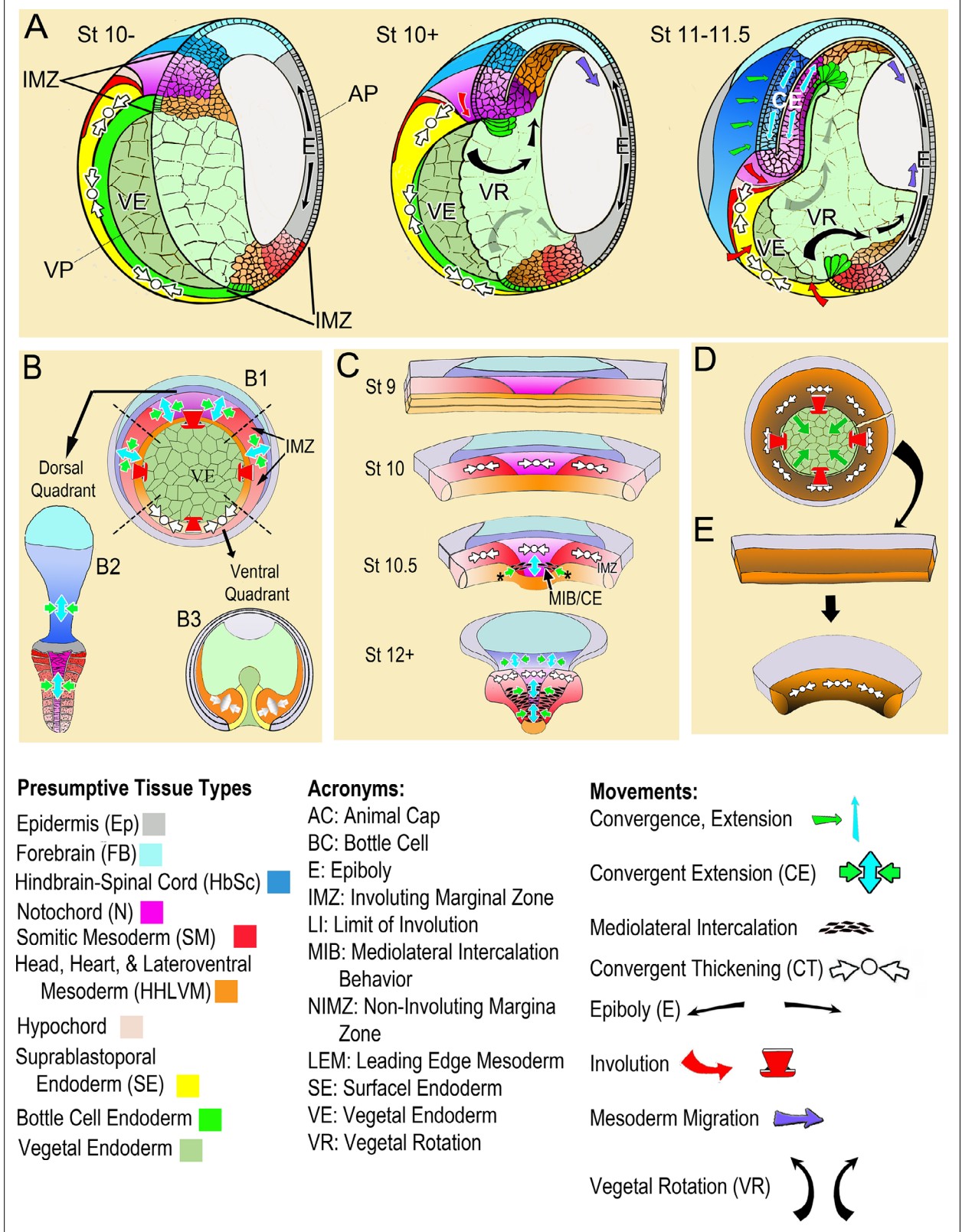

**Figure 1.** Morphogenic movements in the embryo and explants. (**A**) The behavior of the involuting marginal zone (IMZ) in blastopore closure is shown in the pre-gastrula embryo (St 10-) and at subsequent stages of gastrulation (St. 10+, 11–11.5) in mid-sagittal cut-aways rotated 45°, within the context of the whole embryo and the other morphogenic movements. This includes the animal and vegetal poles (AP, VP), the involuting marginal zone (IMZ), the vegetal endoderm (VE), and the morphogenic movements convergent thickening (CT, white arrows), epiboly (E, black arrows within epidermis), vegetal

*Figure 1 continued on next page*

*Figure 1 continued*

rotation (VR, gray arrows/black arrows within VE), mesendoderm migration (purple arrows), involution (red arrows), and convergent extension (CE, green (**C**) and blue (**E**) arrows within neural and mesodermal tissues). See legend of tissue colors, acronyms and movements at the bottom of the figure. The IMZ is a ring of several layers of deep cells, plus a superficial epithelium, that lies above and around the VE prior to gastrulation (St 10-). Deep IMZ cells are presumptive mesoderm whereas the superficial cells are primarily presumptive endoderm, with some presumptive mesoderm dorsally (*Shook et al., 2004*). After involution, the early involuting tissues originating from the lower (vegetal) region of the IMZ, including the presumptive head, heart, and lateroventral mesoderm (orange), migrate across the blastocoel roof (St 10+, 11–11.5, purple arrows), whereas tissues originating from the upper (animal) region of the IMZ, the presumptive notochordal (magenta) and somitic (red) mesoderm engage in CE (green and blue arrows), as does the overlying neural tissue (dark blue). Only the most recently involuted notochordal and somitic tissue, lying just around the inside of the blastopore directly generate convergence forces closing the blastopore by CE. The *non-involuting marginal zone* (NIMZ) is the ring of tissue above the IMZ that bounds the blastopore at the end of gastrulation, with its vegetal edge being the 'Limit of Involution' (LI; see *Figure 1—figure supplement 1B*). The dorsal and dorsolateral sectors of the NIMZ contains the posterior neural tissues (*Figure 1A* St 11–11.5, darker blue cells), which will undergo CE in parallel to the underlying somitic and notochordal mesoderm, driven by an intercalation behavior similar to MIB (*Keller et al., 1992*; *Elul et al., 1997*; *Elul and Keller, 2000*; *Ezin et al., 2003*; *Ezin et al., 2006*). (**B**) A vegetal view of the early gastrula shown without the superficial, epithelial layer over the IMZ reveals the underlying deep presumptive mesoderm (**B1**). Sandwich explants of the dorsal quadrant of the Marginal Zone (IMZ plus NIMZ, **B2**), including tissues of presumptive notochordal (magenta), somitic (red), and neural (blue) fates undergo CE (green and blue symbols)(for detailed anatomy of explants, see *Figure 1—figure supplement 1*). In contrast, explants of the ventral quadrant (**B3**) develop only ventrolateral mesoderm (orange) when separated from the organizer ('dorsal') side of the embryo, round up and show a thickening of the mesodermal tissues, a movement called Convergent Thickening (CT, white symbols). Together, CE on the dorsal side and CT on the ventral side of the IMZ were previously thought to provide the convergence forces that close the blastopore in the intact embryo (*Keller and Danilchik, 1988*) (**B1**). (**C**) In the current work, a more detailed analysis of 'giant' sandwich explants of the entire Marginal Zone, and variants thereof (*Figure 1—figure supplement 1A*), reveals a revised view of the pattern of CT expression, wherein the entire IMZ uniformly expresses CT during early gastrulation (white symbols, St 10). At the midgastrula stage, expression of CT begins to transition into a progressive expression of CE (green and blue arrows, St 10.5), a post-involution progression from anterior to posterior, as described previously (*Shih and Keller, 1992a*), while CT continues in the pre-involution IMZ; CE also begins in the posterior neural tissue (St 10.5–12+). (**D**) Lastly, ventralized embryos, which lack the presumptive tissues of notochordal and somitic mesoderm and contain only presumptive ventrolateral tissues, involute their marginal zone (red arrows) and close their blastopore symmetrically (green arrows), using CT alone (white symbols), and giant explants of ventralized embryos (**E**), consisting entirely of ventral tissues, undergo uniform CT. For a complete description of these and other types of explants, see Materials and methods and *Figure 1—figure supplement 1*.

The online version of this article includes the following figure supplement(s) for figure 1:

**Figure supplement 1.** Description of explant construction.

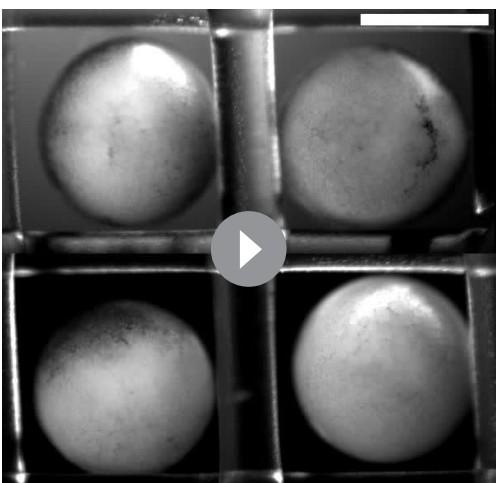

*2004*; *del Pino et al., 2007*; *Moya et al., 2007*; *Venegas-Ferrín et al., 2010*; *Elinson and del Pino, 2012*) suggest that CT may be a general mechanism that functions throughout the amphibians. In the case of *Gastrotheca riobambae*, CE is delayed until after blastopore closure and gene markers for tissues known to undergo CE are not expressed until neurulation (*del Pino and Elinson, 1983*; *del Pino, 1996*; *Benítez and Del Pino, 2002*). These observations suggest that such embryos rely primarily on CT for blastopore closure (Shook, personal observations, *del Pino and Elinson, 1983*; *del Pino, 1996*; *Benítez and Del Pino, 2002*; *Keller and Shook, 2004*; *del Pino et al., 2007*; *Elinson and del Pino, 2012*), that CT is a widely used convergence machine independent of CE, and that CT warrants further investigation as an independent movement.

Here, we show that CT is driven by forces resulting from increased interfacial tissue surface tension between the deep mesoderm and its overlying superficial epithelium, a mechanism fundamentally different from that driving CE. We confirm that CT begins uniformly throughout the IMZ from the onset of gastrulation and continues to operate in the pre-involution IMZ throughout gastrulation (*Figure 1A and C*). We show that

**Video 1.** Blastopore closure in normal vs. ventralized embryos. Top two embryos are normal, bottom two were UV ventralized. The two on the left are those shown in Figure 6A. Stage 10–20 (G + 0.25 h to G + 12.75 h); most of neurulation is occurring on the opposite side of the normal embryos. 3 min/frame. Scale bar = 1 mm.
https://elifesciences.org/articles/57642/figures#video1

presumptive notochordal and somitic tissues expressing CT undergo an anterior-to-posteriorly progressive transition from expression of CT to CE as they involute (*Figure 1A and C*). Thus we demonstrate that CT acts outside the blastopore, in parallel with CE acting inside the blastopore, to generate forces driving blastopore closure and the internalization of the presumptive mesodermal and endodermal tissues. We show that the underlying mechanism driving CT is distinct from that driving CE. Lastly, we summarize a working model for the mechanism of CT and its integration with CE and other morphogenic movements in *Xenopus*.

CT represents a novel but likely common mechanism of using isotropic change in IFT to generate tensile force, which within the geometric context of the tissue, results in oriented tension, producing polarized, anisotropic tissue movements. Taken together with evidence in the literature (*Keller and Shook, 2004*; *del Pino et al., 2007*; *Shook et al., 2018*), our findings suggest that CT is a common feature of anuran gastrulation, with CE beginning at different times during or after gastrulation in different species. The significance of these results for the use of tissue surface tension-based mechanisms in the morphogenesis of other organisms and for the evolution of amphibian and chordate gastrulation generally are discussed.

## Results

Note that, rather than being mere supplemental materials, our movies are, in many cases, the clearest, most compelling illustration of our results. The most important ones are marked with a "*" appended to the Video number.

### CT is autonomous, begins prior to CE and acts throughout the pre-involution IMZ

Tissue explants allow expression of region specific morphogenic movements free of the mechanical interactions and constraints within the intact embryo and thereby distinguish active, autonomous local processes from passive responses due to mechanical linkages with other tissues. To test the autonomy and dynamics of CT, we isolated the entire marginal zone (both *non-involuting marginal zone* (NIMZ) and IMZ) including bottle cells plus some vegetal endoderm from Stage 9 embryos as 'Standard Giant' sandwich explants (*Figure 1—figure supplement 1A1,B,C*). We also made identical explants but without bottle cells (*Figure 1—figure supplement 1A2*) to assay whether the tension generating properties of bottle cells (see *Hardin and Keller, 1988*) contribute significantly to CT. Both types of explants show strong, uniform convergence of the IMZ (*Figure 2A*, *Figure 2—figure supplement 1A*; IMZ = lower, generally less densely pigmented region below dashed yellow line; *Video 2*\*) through Stage 10.5, 2 hr after the onset of gastrulation (G + 2 h), after which the transition to CE begins. Ventralized Giant sandwich explants made from ventralized embryos (*Figure 1D and E*; *Figure 1—figure supplement 1A6*) also show uniform convergence of the IMZ but do not transition to CE (*Figure 2B*, *Video 3*\*).

Plots of regional convergence of the widest portion of the IMZ across its mediolateral extent in standard (dorsally-centered) giant explants (*Figure 1—figure supplement 1A1*) and also in ventrally-centered giant explants (*Figure 1—figure supplement 1A4*), which control for healing artifacts at the lateral edges, show no mediolateral differences in convergence prior to the onset of CE at G + 2 h (*Figure 2C and D*, *Video 4*). The rates of convergence in these explants through G + 2 h were similar to those observed in giants made from ventralized embryos (*Figure 1—figure supplement 1A6*), which show continuous CT of the IMZ for several hours after explantation (*Figure 2B and E*; *Video 3*). Likewise, giant sandwich explants made from Li + dorsoanteriorized embryos (*Kao et al., 1986*; *Kao and Elinson, 1988*; *Kao and Elinson, 1989*; *Figure 1—figure supplement 1A7*) show isotropic convergence (CT) before they extend, and also after they begin to extend, as all sectors of these embryos are 'dorsal' and undergo CE (see refs cited above) (*Figure 2—figure supplement 1B*; *Video 5*).

In sandwich explants including dorsal tissue made at G-2 to –3 h, 2–3 hr prior to the traditional beginning of gastrulation at Stage 10 (G0), with the dorsal midline identified by 'tipping and marking' the embryo during the one-cell stage (see Materials and methods), the IMZ and NIMZ regions initially converged at the same rate when the cover-glass was removed after healing, but the IMZ began to consistently show >4%/hour higher convergence than the NIMZ around Stage 10- (G-0.7h, SEM =

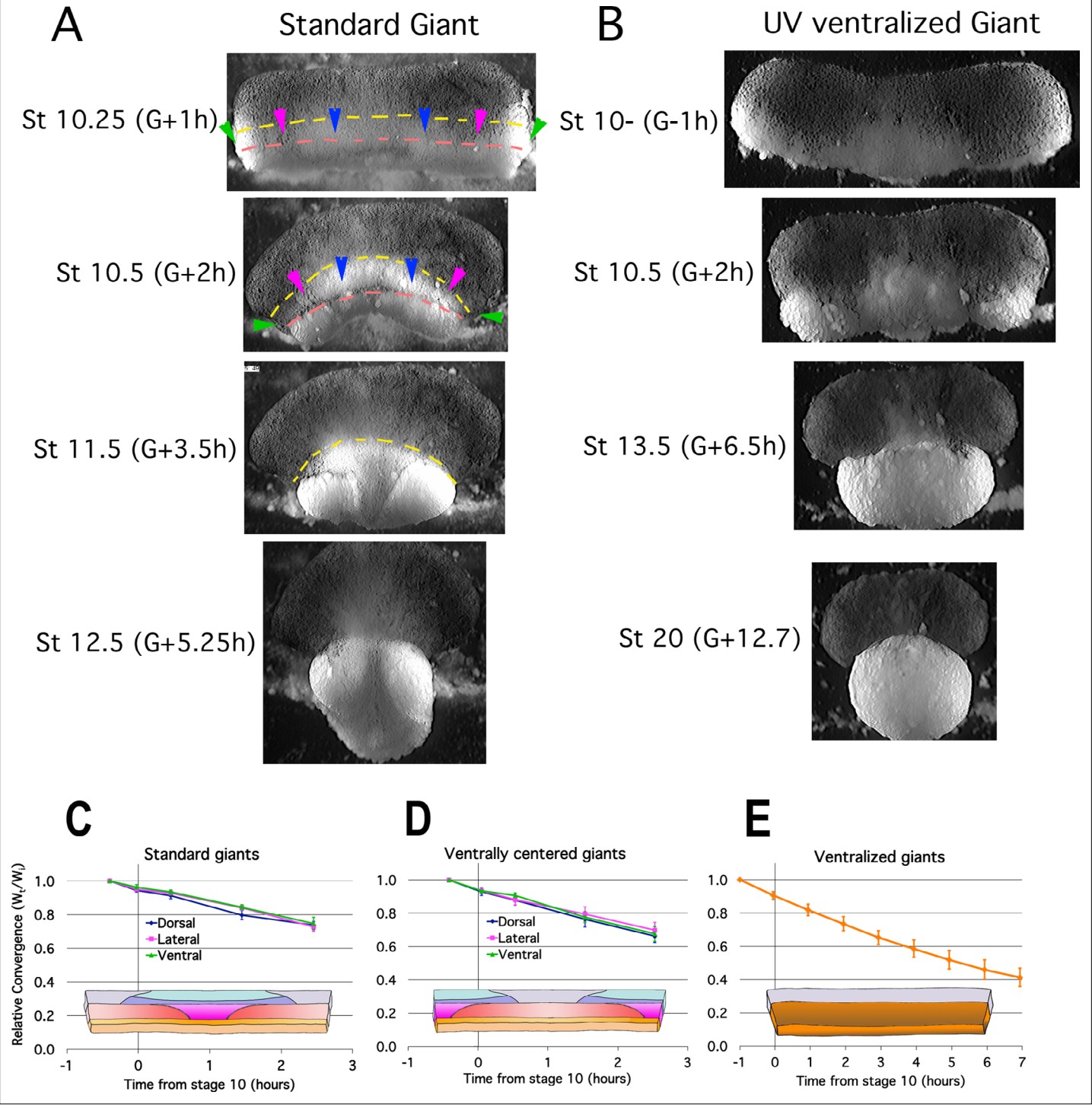

**Figure 2.** Early Convergence is uniform across regions. Frames from time lapse movies of standard giant sandwich explants (**A**; *Video 2**, left) or ventralized standard giants (**B**; *Video 3**) after release from their constraining coverslips show uniform convergence of the IMZ. The upper limits of the upper and lower IMZ are indicated by dashed yellow and pink lines, respectively (**A**). To compare regional rates of convergence in such explants, the mediolateral extent of the dorsal (A, between blue pointers), lateral (between blue and magenta pointers), and ventral regions (between magenta and green pointers) of the IMZ were measured at times from Stage 10- and plotted with respect to initial width ($W_t/W_i$) in standard (dorsally centered) giants (n = 3–5 embryos per time point) (**C**) and ventrally centered giants (n = 4–5 embryos) (**D**). The convergence of the entire IMZ was also measured in giants from ventralized embryos (n = 3 embryos) (**E**); error bars = SEM. For C-E, the same explants were measured at sequential time points.

The online version of this article includes the following source data and figure supplement(s) for figure 2:

*Figure 2 continued on next page*

*Figure 2 continued*

**Source data 1.** *Figure 2C, D* source data: Relative live explant convergence vs. time.

**Figure supplement 1.** CT in giants without bottle cells and in Li⁺ Dorsoanteriorized giant sandwich explants.

+/- 0.2 h, n = 4), such that explants began to bow toward their vegetal (lower) edge (For example, see first 10 frames vs. subsequent frames of *Videos 6 and 7*). More specifically, the IMZ region converges at about 10% per hour both before and after Stage 10-, in agreement with prior results (*Shook et al., 2018*), whereas the NIMZ region converges at about 7% per hour earlier (not significantly different from IMZ, p = 0.12, n's = 4 vs 4) but then slows to an average of 0% per hour over the subsequent hour (significantly slower than IMZ, p < 0.001, n's = 5 vs 5) (see *Source data 1* ), and does not begin to converge again prior to G + 2 h (mean convergence = –0.4% /hr, SEM = 0.9, between G-0.7 and +1.9 h; n = 4). Notably, the onset of this differential convergence begins at about the same time in both normal and ventralized sandwich explants (for example of ventralized sandwich, see first 10 frames vs. subsequent frames of *Video 3*). Note that the initial, pre-Stage 10- convergence of the NIMZ should not be confused with the CE of the presumptive posterior neural tissue in the dorsal region of the NIMZ at the mid-gastrula stage (see *Keller and Danilchik, 1988*; *Keller et al., 1992*). The onset of differential NIMZ and IMZ convergence is one of several behaviors associated with the onset of CT (see Discussion).

These results indicate that the entire IMZ undergoes CT prior to the onset of CE without dorsoventral bias in timing or degree, and that early convergence is independent of any convergence forces from bottle cell formation (see Discussion). Also, CT occurs in the pre-involution IMZ, regardless of whether CE occurs later in its normal pattern, does not occur at all (as in ventralized embryos), or occurs later across the entire IMZ (as in dorsoanteriorized embryos). These results establish that CT is an intrinsic property of the IMZ at the onset of gastrulation and is independent of CE, and they are consistent with the idea that CT is expressed in the same pattern at the onset of gastrulation in normal, ventralized and dorsoanteriorized embryos, suggesting CT is also independent of dorsal-ventral patterning.

## CT is a specific property of the IMZ

To compare CT of the NIMZ vs. IMZ, we isolated the two regions, using explants designed to reduce potential artifacts (see below). By measuring convergence and thickening over time in live explants, we were limited to measuring profiles, but this reduced the significant problem of embryo-to-embryo variability and variations in explant construction that we observed when we attempted to make these measurements on explants that had been aged and fixed at different time points (e.g. *Figure 3— figure supplement 1A-D*). To measure CT alone and avoid the confounding effects of the progressive transition to CE in the dorsal IMZ and the onset of neural CE of the dorsal NIMZ (presumptive neural), we made sandwich explants of the ventral 180° of the embryo (V180s; *Figure 1—figure supplement 1A5*; e.g. *Video 8*) before Stage 10+; when made during early gastrulation, these explants, like explants from ventralized embryos, rarely differentiate dorsal tissues and remain ventral in character (personal observations and *Dale and Slack, 1987*). IMZ and NIMZ regions were cut from V180 explants and used to make 'double' ventral IMZ and NIMZ

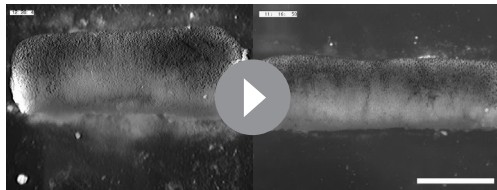

**Video 2.** Normal giants with and without bottle cells. Giant with bottle cells on left, without on right. Movies begins about Stage 10.25 (G + 30 m), just after release from cover glass. 3 min per frame. Total Elapsed time = 10:15. Time stamp on right movie is fast by 1:02 (h:mm). Scale bar = 1 mm.
https://elifesciences.org/articles/57642/figures#video2

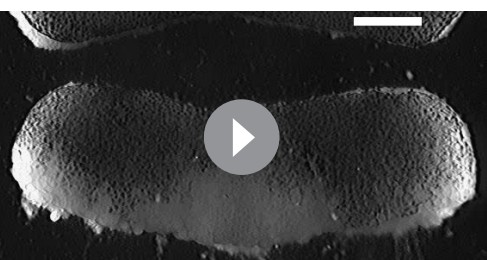

**Video 3.** Early UV ventralized giant. Movie begins at control Stage 9.5 (G-1h). 3 min per frame. Total elapsed time = 14:52. Scale bar = 500 µm.
https://elifesciences.org/articles/57642/figures#video3

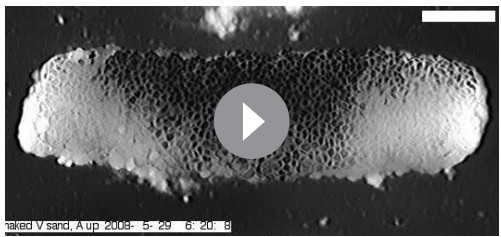

**Video 4.** Ventral giant sandwich. Made from normal embryos, cut through the dorsal midline. Only a few bottle cells included. NIMZ and IMZ regions converge equally through frame 10 (G-1h), after which IMZ converges more rapidly. Begins Stage 9+ (G-1.5h) through early neurulation (G + 7.45). 3 min per frame, 180 frames, 15 fps. Scale bar = 500 μm.
https://elifesciences.org/articles/57642/figures#video4

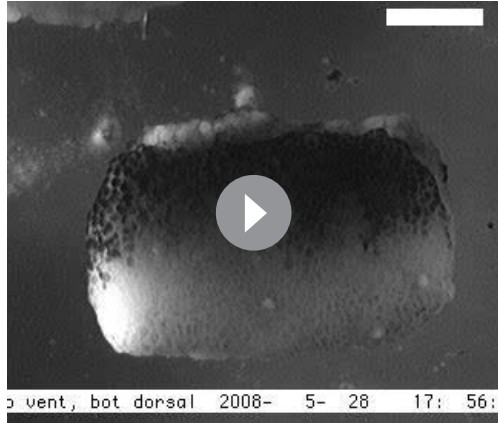

**Video 6.** DMZ 120° sandwich explant showing bottle cell retraction and spreading. Note also adjacent epithelial are pulled toward forming bottle cells. Starts Stage 9+ (G-1.75h) through early neurulation (G + 7.75 h). 3 min/frame. Scale bar = 500 μm.
https://elifesciences.org/articles/57642/figures#video6

sandwich explants (*Figure 1—figure supplement 1A8*), to isolate each region from the other and to avoid artifacts resulting from prolonged healing. We then imaged these double explants from above and the side, respectively (*Figure 3A, B*; e.g. *Video 9**, 10*).

The behavior of the IMZ and NIMZ differ dramatically in several regards. First, double ventral IMZ sandwich explants show more rapid and more extensive convergence than double ventral NIMZ explants (*Figure 3A and C*; *Video 9**), explaining the initial bowing of giant explants to form a concave vegetal edge (e.g. *Figure 2A*, St 10.5). This is due to the mechanical linkage of laggard NIMZ with the more rapidly converging IMZ. Second, the superficial epithelial layer that initially covers the deep regions of double IMZs retracts to a small area (*Figure 3A*, pointers; *Video 9**), which leaves most of the deep region uncovered, a behavior associated with CT of the IMZ (see below) but one that does not occur in the double NIMZs. Lastly, convergence of the IMZ explants is accompanied by thickening, whereas NIMZ explants show no thickening (*Figure 3B and D*; *Video 10**).

Convergence of Double IMZ explants showed an initial rapid phase (*Figure 3C*, G + 2–3 h), which correlated with the rapid thickening of the IMZ observed in the first hour after release of the initial VMZ sandwich explants from the cover glass (data not shown), during the lateral healing together of the two IMZs (see *Figure 1—figure supplement 1A8*). A slower convergence phase followed, which correlated with the continuing, slower

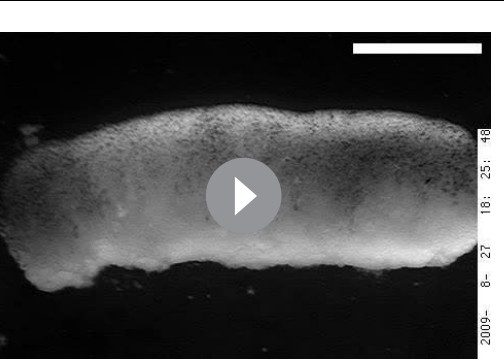

**Video 5.** Lithium dorsoanteriorized giant. Movie begins at control Stage 10+ (G + 0.5 h). 3 min per frame. Elapsed time to penultimate frame = 15 h; final frame is at 17 hr. Scale bar = 1 mm.
https://elifesciences.org/articles/57642/figures#video5

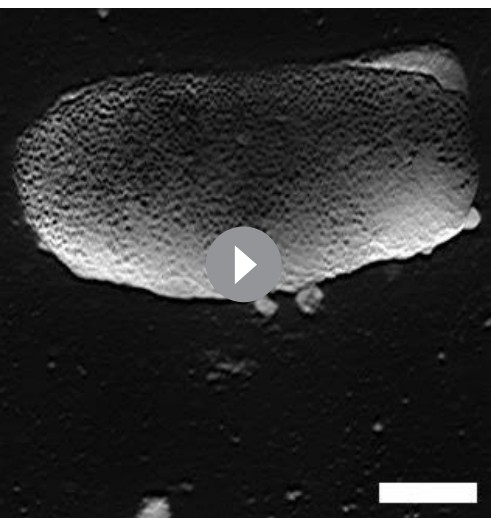

**Video 7.** DMZ 180° sandwich explant from Stage 9.5–16 (G-1h to G + 9 h). 3 min/frame, scale bar = 500 μm.
https://elifesciences.org/articles/57642/figures#video7

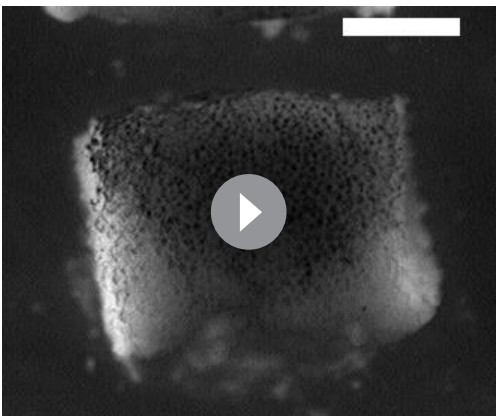

**Video 8.** Ventral 180° sandwich explant. Movie begins at control Stage 9.5 (G-0.9h). 3 min per frame. Elapsed time = 8 hr. Scale bar = 500 μm.
https://elifesciences.org/articles/57642/figures#video8

thickening of the Double IMZ explants (*Figure 3C and D*, G + 3 h onward). In contrast, Double NIMZ explants showed only slow mediolateral convergence, and did not thicken (*Figure 3C and D*), but it did slowly elongate orthogonally within the horizontal plane, that is along the initially shorter animal-vegetal axis, making the explant more circular (*Figure 3A*, *Video 9*). We interpret this gradual circularization of the NIMZ as the general tendency for isolated tissue to round up, coupled with the cell behaviors underlying epiboly causing the tissue to remain thin. Neither the Double IMZ nor NIMZ explants showed apparent differences in convergence or thickening in the animal vs. vegetal edges within the tissue, illustrating that CT does not show the patterned anterior-to-posterior progression characteristic of CE (*Shih and Keller, 1992b*; *Lane and Keller, 1997*). These results demonstrate the distinct properties of IMZ and NIMZ tissues.

In intact embryos, the pre-involution IMZ does not thicken during early gastrulation but instead continues to thin (*Figure 3—figure supplement 2A, B*), as does the NIMZ (*Keller, 1980*). Thickening of the IMZ is prevented by the mechanical resistance to its convergence by adjacent tissues, notably the vegetal endoderm (See Discussion).

## Deep IMZ cells lose their affinity for superficial epithelial cells at the onset of CT

The retraction of the superficial epithelium from deep IMZ but not from deep NIMZ cells in sandwich explants (*Figures 2A and 3A*) suggests a loss of affinity between deep and superficial IMZ tissues during CT. We use 'affinity' here in the broader, Holtfreterian sense of dynamic, regulated cell-cell contact behavior (*Holtfreter, 1939*; *Townes and Holtfreter, 1955*), rather than in the Steinbergian sense of adhesive competition (*Steinberg, 1964*; *Steinberg, 1970*). As a measure of deep-superficial tissue affinity, we assayed the spreading ability of the epithelial layer by culturing it on explants of contiguous deep IMZ and NIMZ beginning at Stage 8.5–9 (G-3 to –2h) embryos, on a fibronectin (FN) substrate, whiandch stabilizes the explant. A strip of epithelial tissue was placed medially, deep side down, across the entire animal-vegetal extent of the explant, spanning the IMZ-NIMZ boundary, and covered with a cover-glass to promote attachment (*Figure 4A*). After attachment, the cover-glass was removed, and the behavior of the strip over the lower (vegetal) IMZ, the upper (animal) IMZ, and the NIMZ regions was recorded over time (regions indicated in *Figure 1—figure supplement 1*B and *Figure 2A*). Beginning around Stage 10-, the epithelial tissue lying over the lower deep IMZ began to retract and/or self-adhere (basal to basal; see *Luu et al., 2011*), reducing its contact area, whereas the epithelium over the deep AC-NIMZ region remained stable or spread further over time (*Figure 4B and C*; *Video 11**). High resolution movies suggest that epithelium over the upper deep IMZ also reduces its area (*Video 11**), but in most cases there is so much movement of the adjacent, visible deep cells, it is difficult to reliably determine exactly where the interface between the deep upper IMZ and NIMZ regions lies, so these results were not included in our analysis. FN attachment stabilizes the explant such that it does not roll over once the cover glass is removed. Although FN could alter the character of the opposite, assay side of the deep tissues and generate artifacts, the same results were obtained when we cultured explants on Poly-D-lysine coated substrates (data not shown). Notably, transplanted epithelium over deep ectoderm does not spread until Stage 10- (*Figure 4C*); the same is true of endogenous epithelium adjacent to naked deep ectoderm (DRS, personal observations).

In similar experiments, the deep tissue was cultured on a BSA coated substrate (which blocks deep cell adhesion to the substrate), with patches of epithelium spanning the IMZ-NIMZ boundary. To prevent the explant from rolling and to hold the epithelium against the deep tissue, while allowing the cells on the upper surface of the explant to rearrange, a thin layer of agarose and a coverslip were

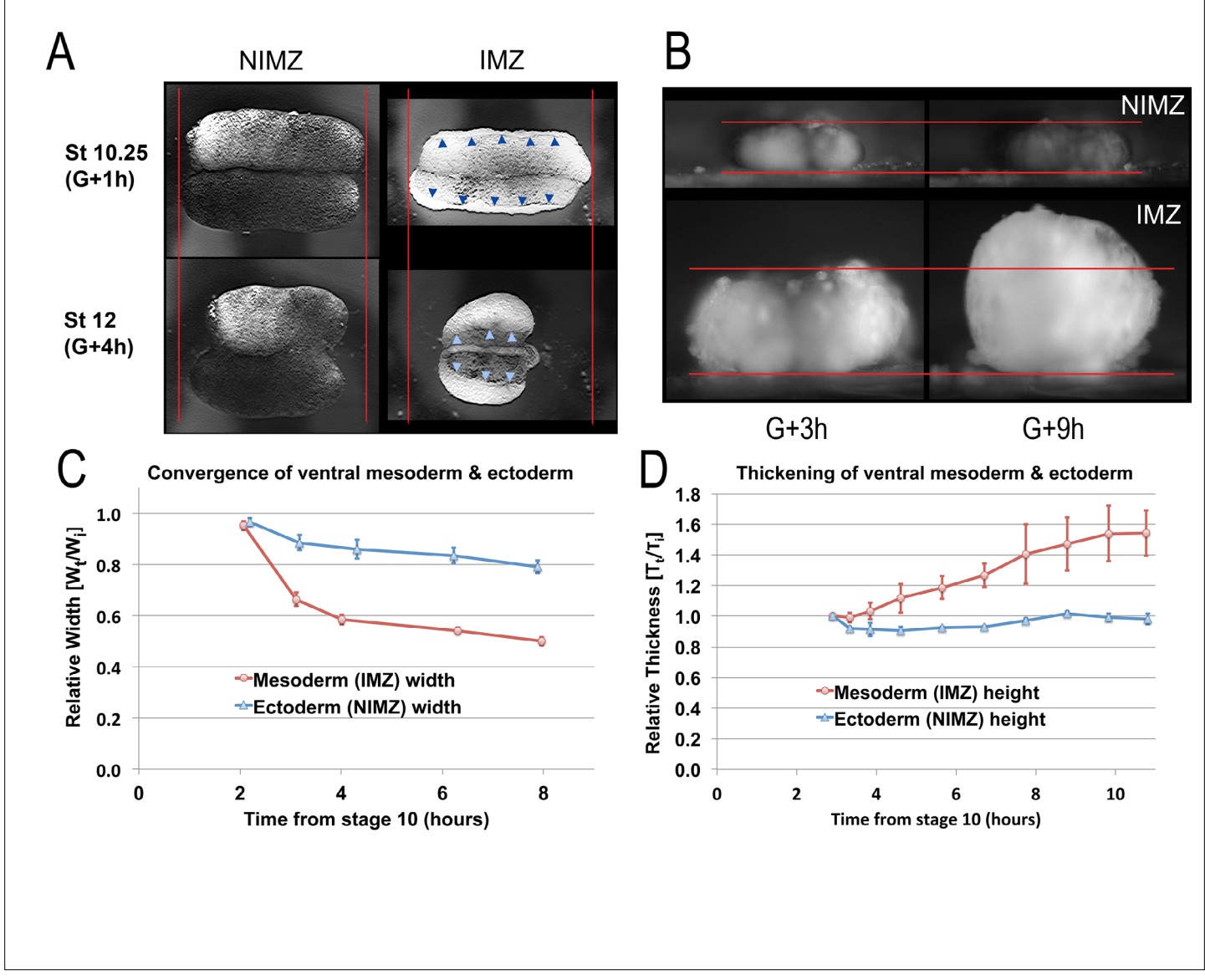

**Figure 3.** Quantitation of convergence and thickening in IMZ vs. NIMZ regions of explants. Double ventral NIMZ and IMZ sandwich explants (see *Figure 1—figure supplement 1A8*) were used to examine the autonomous rates of convergence and thickening in each region, by imaging both from above (**A**, *Video 9**) and from the side, viewing the lateral edge (**B**, *Video 10**); pointers in A indicate the edge of the superficial epithelium on the IMZ explant, red lines in A and B indicate the initial width or height of the explants. Explant width (**C**; n = 6 explants per time point) and thickness (**D**; n = 2 explants per time point) were measured over time ($W_t$ and $T_t$ respectively; see Materials and methods for details) and plotted with respect to the initial value ($W_i$ and $T_i$); error bars = SEM. Measures for C and D were made from the same explants at sequential time points.

The online version of this article includes the following source data and figure supplement(s) for figure 3:

**Source data 1.** *Figure 3C, D* source data: Relative live explant thickness and width vs. time.

**Figure supplement 1.** Giant sandwich explants were made from rhodamine dextran amine (RDA) labeled embryos (**A,B**) prior to the onset of gastrulation, allowed to heal under a cover glass for 30 minutes, then allowed to equilibrate for at least 1 hour after cover glass removal, prior to fixation.

**Figure supplement 1—source data 1.** *Figure 3—figure supplement 1C, D* source data: Fixed explant thickness vs. time.

**Figure supplement 2.** LSCM images of RDA injected embryos (**A**) were used to measure regional thickness of the IMZ and NIMZ, before and during early gastrulation.

**Figure supplement 2—source data 1.** *Figure 3—figure supplement 2B* source data: Projected sandwhich explant thickness.

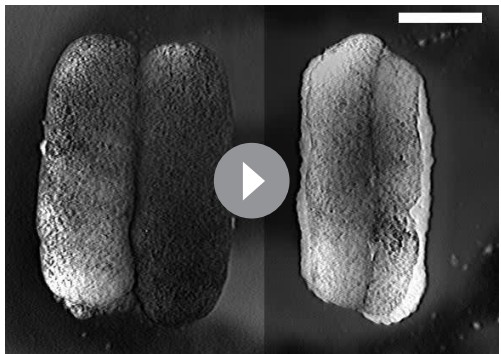

**Video 9.** Double NIMZ and double IMZ sandwich explants. Movie begins at G + 1 h. 3 min/frame. Elapsed time 8.9 hr. Scale bar = 500 μm.
https://elifesciences.org/articles/57642/figures#video9

lightly pressed against the explant throughout the experiment (*Figure 4D*). Under these conditions, the epithelium contracted and retreated from the surface of the deep IMZ onto the surface of the deep NIMZ, where it re-spread (*Figure 4E*); note that the deep cells of the IMZ spread upward and over the deep NIMZ cells (*Figure 4E*), such that they appear to follow or push the epithelium as it retreats animally, a behavior more easily appreciated in the time lapse recording (*Video 12*\*). Comparing the G + 1 h to the G + 6 h stills (*Figure 4E*), you can see that the epithelium has actually moved significantly with respect to the initial boundary, which it overlapped. Looked at another way, the deep mesodermal cells compete with the epithelial cells for adhesion to the deep ectoderm. One notable difference between the two variants of this experiment is that the uncovered epithelium is free to self-adhere basally (*Figure 4A and B*), whereas when under glass, the basal surface remains pressed against the deep tissue, thus allowing for re-spreading on the NIMZ (*Figure 4D and E*).

These results support the idea that the affinity between superficial epithelium and IMZ cells decreases at Stage 10- (G-0.7h), while that between epithelium and NIMZ cells increases. The changes in spreading or contraction behavior occur regardless of the region of the embryo from which the epithelium was taken (e.g. AC/NIMZ vs. IMZ superficial epithelium), which strongly suggests that it is the deep IMZ tissue that lowers its affinity for the overlying epithelium around the beginning of gastrulation (Stage 10, G0), rather than the epithelium changing its affinity for the deep IMZ.

## Interfacial tension and tissue surface tension in the IMZ increase during CT

Tissues, like other types of soft matter, can be characterized by material properties such as tissue surface tension (TST), which is a measure of the cohesion of liquid-like tissues. In physical terms, TST is represented by a surface energy per unit area, which acts to minimize the tissue–medium interface. A classic example of TST is demonstrated when tissue explants with initially irregular shapes 'round-up' over time until they are transformed into spherical aggregates. Similarly, the interfacial tension (IFT) acts to minimize the interface area between two non-intermixing tissues. Our results show that in contrast to the NIMZ, the IMZ develops an increasing tendency to round up (*Figure 3A and B*) and a reduced affinity for superficial epithelial layers (*Figure 4*). This supports the hypothesis that CT is the result of an increase in the IFT between the deep IMZ and its overlying epithelium, associated with the decrease in affinity between the two. Such an increased IFT could drive the observed convergence and thickening of tissue explants and force generation along the long (mediolateral) axis as the deep and superficial layers tend to minimize their interfacial surface area (see Discussion).

To test the idea that IFT increases around G0 we used a parallel plate compression device (the MicroSquisher; see *Figure 5—figure supplement 1*, Materials and methods for details). We first tested a range of compressive strains on explants of deep mesodermal or ectodermal tissue wrapped in large pieces of superficial epithelium. The final configuration of such explants resembled a 'wonton' (as in *Figure 5A*) and were thus called wonton explants (see Materials and methods). An excess of epithelium ensured that pseudo-elastic stretching of this layer did not contribute to tensions at the wonton surface (*Luu et al., 2011*), such that only the IFT of the deep tissue affects

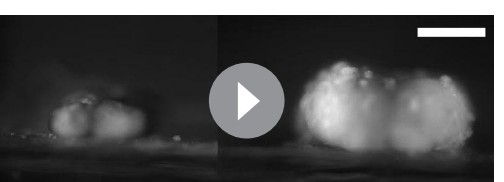

**Video 10.** Double ventral IMZ and NIMZ explants viewed from the side, facing lateral ends. Movie begins at about G + 3 h. 5 min/frame. Elapsed time = 6 hr. Scale bar = 500 μm.
https://elifesciences.org/articles/57642/figures#video10

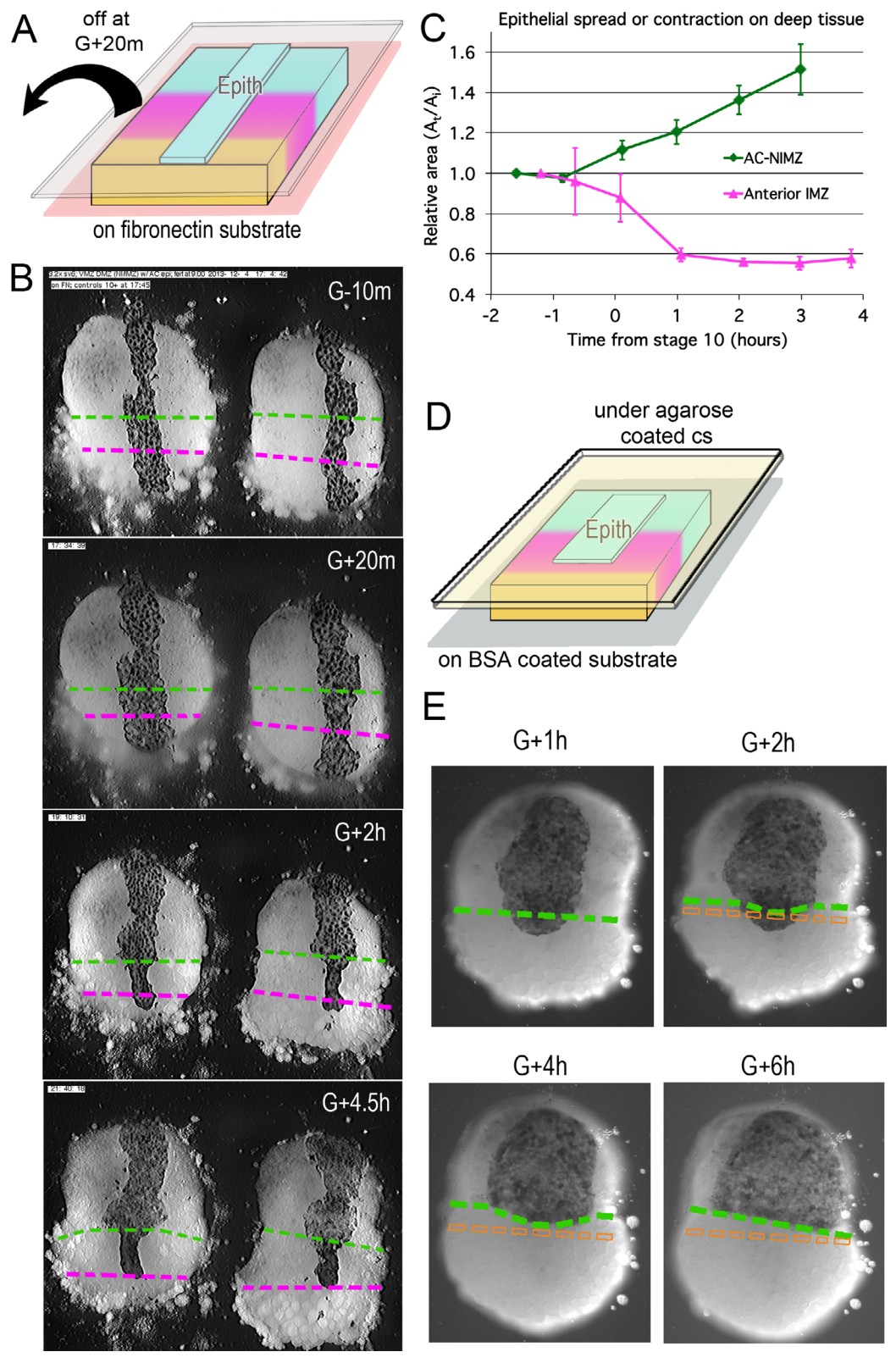

**Figure 4.** Epithelial affinity. (**A**) Explants of deep IMZ plus NIMZ tissue were made prior to the onset of gastrulation and stabilized by adhering them to fibronectin-coated cover glass below the explant; strips of AC epithelium large enough to span the entire animal-vegetal extent of the deep tissue were subsequently grafted onto its surface and kept in place with cover glass above the explant. (**B**) In the examples shown (lower IMZ below magenta dashed

*Figure 4 continued on next page*

*Figure 4 continued*

line, upper IMZ between magenta and green, NIMZ & AC above green dashed line; boundaries between the three regions identified by cell, motility, size and gray cast), the movie (*Video 11\**, where these events can be most easily seen) begins 10 min prior to the onset of gastrulation (G-10m). The epithelium adhered and spread over all regions (G + 20 m), after which the upper cover glass was removed. Immediately afterward, epithelium over both NIMZ and IMZ regions contracted slightly, then that over the NIMZ re-spread, while that over the IMZ contracted more strongly and continuously, often to the point of becoming detached from the deep tissue, and in many cases retracted from the lower (vegetal) edge of the deep tissue (G + 2 h). Tissue over the IMZ remained contracted, while that over the NIMZ tended to spread (G + 4.5 h). (**C**) The relative area of spreading or contraction of the epithelial tissue over different deep tissue regions over time was quantitated, beginning after cover glass removal (each time point based on measures from 2 to 8 explants at sequential time points; error bars = SEM). (**D**) In a variation on this assay, explants of deep ventral IMZ plus NIMZ tissue were cultured on BSA coated cover glass (below) and patches of epithelium were lapped across the NIMZ/IMZ boundary and held apposed to the deep tissue with an agarose coated cover glass (above) throughout the experiment. In the example (**E**), the epithelium initially over the deep IMZ (below green line) first contracts animally (G + 2 h), eventually retracting across the boundary (G + 4 h), then respreads on the deep NIMZ (G + 6 h) (*Video 12\**, where these events can be most easily seen). During this time, upper deep IMZ cells (mesoderm) spread over deep NIMZ cells (ectoderm), such that the NIMZ/IMZ boundary on the surface of the deep tissue is shifted from its original position (dashed orange line). The deep NIMZ/IMZ boundary is based on differential cell size and motility, which are evident in the movie.

The online version of this article includes the following source data for figure 4:

**Source data 1.** *Figure 4C* source data: area of epithelia ove IMZ and NIMZ vs. time.

the overall shape of the wonton. Sandwiches of AC tissue (superficial and deep presumptive ecto-derm) behave indistinguishably from wonton explants containing deep AC tissue and are much easier to make at the desired stage, so were also used to test IFT; the epithelial surface of the AC sandwich explants completely healed over the deep tissue by Stage 10. Strains of 10–30% gave consistent results at a given strain, but we found that for explants of AC tissue, IFT increased suddenly above a strain of 22%, and a similar increase was seen for IMZ explants (*Figure 5—figure supplement 2C,D*), suggesting solid-like behavior at high strains. Strains of 18–22% were therefore used in our experiments. We saw increasing IFT

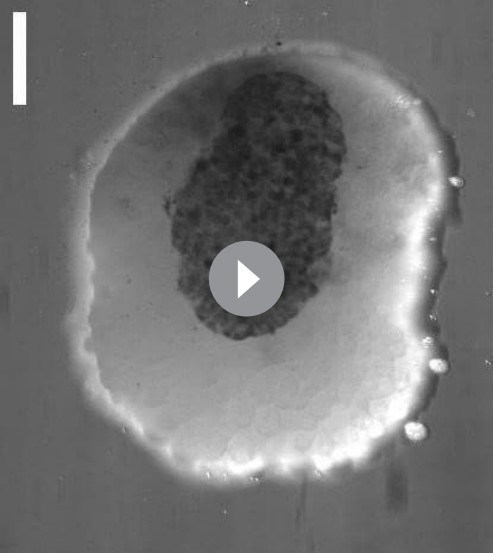

**Video 12.** Epithelial retreat and respreading. Movie from which Figure 4E was made. Animal cap epithelium, recombined with ventral deep MZ. Under cover glass the entire time. Note that deep IMZ cells spread animally over the deep NIMZ tissue between frames 1 and 30 (first 20% of the movie), most obviously at the lateral edges; in response the more vegetal superficial cells over the IMZ initially contract; during the next 20% of the movie, the superficial cells move animally and begin to respread over the NIMZ. Stage 10.25–15 (G + 1 h to G + 8.5 h). 3 min/frame. Scale bar = 100 µm.

https://elifesciences.org/articles/57642/figures#video12

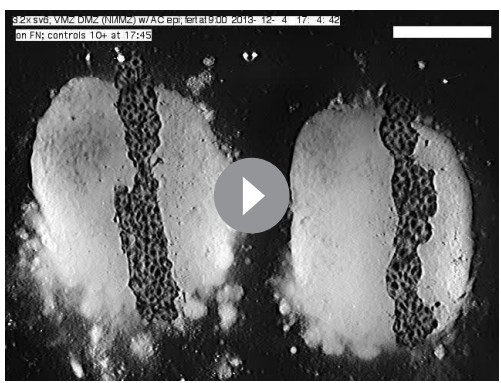

**Video 11.** Epithelial spreading/contraction assay. Movie from which Figure 4B was made. Animal cap epithelium, recombined with deep MZ, ventral on the left, dorsal on the right, on FN coated substrate; vegetal end down. Stage 10–12 (G-0.2h to G + 4.6 h). Cover glass removed between frame 11 and 12. 3 min/frame. Scale bar = 500 µm.

https://elifesciences.org/articles/57642/figures#video11

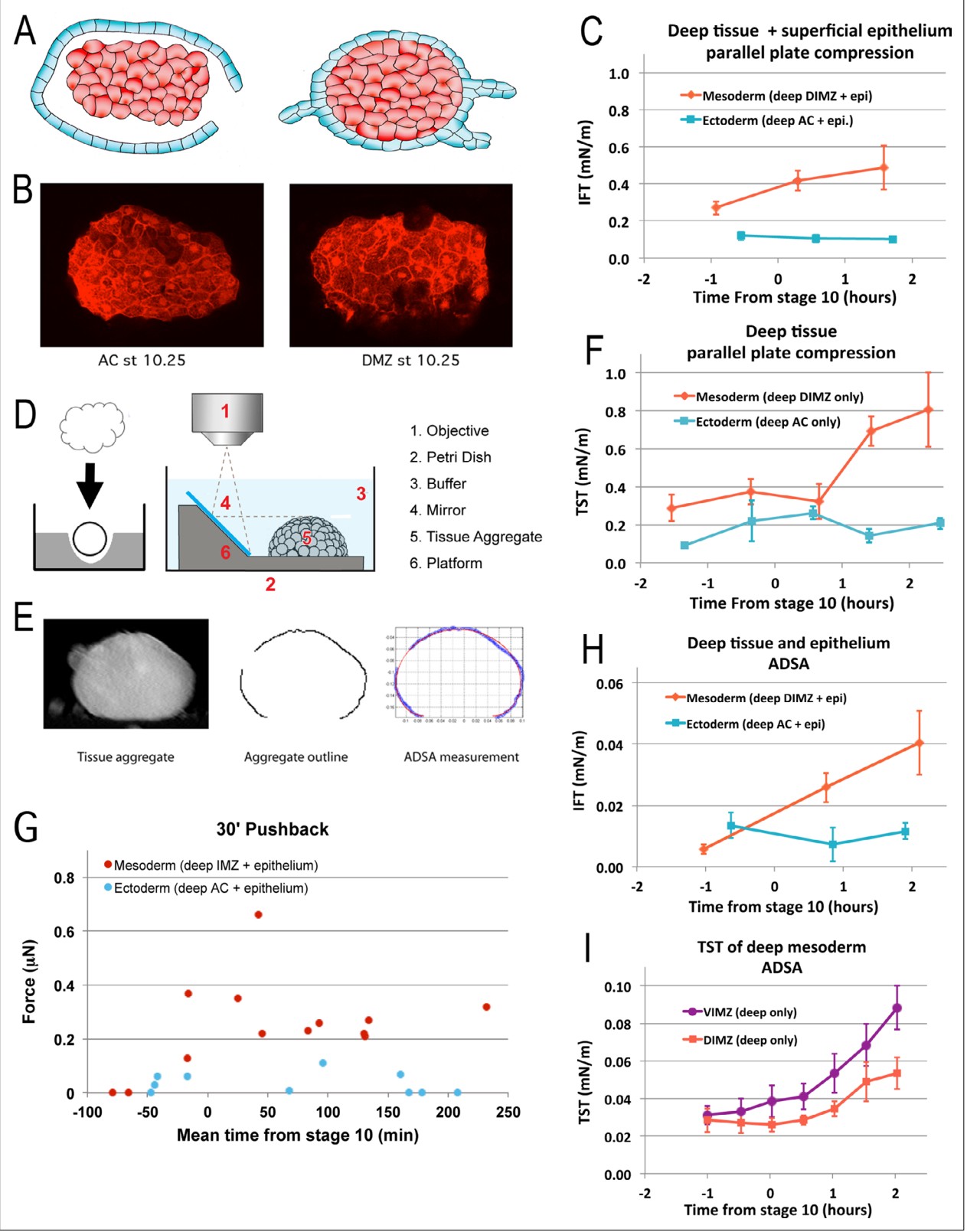

**Figure 5.** Tissue surface tension and interfacial tension rise in association with the onset of CT. Aggregates of deep tissue wrapped in excess superficial epithelium ('wontons'), (**A**), were used to measure the IFT between DIMZ (Mesoderm) or AC (Ectoderm) deep tissue and epithelium (**C**), assayed by parallel plate compression (using a MicroSquisher; see Materials and methods and *Figure 5—figure supplement 1*). Aggregates of deep mesodermal or ectodermal tissue, prepared by explanting deep tissue into an agarose well and allowing the explant to round up (D, left panel) were similarly used

*Figure 5 continued on next page*

*Figure 5 continued*

to measure TST (**F**). IFT and TST were determined using outline and analysis features of the MicroSquisher software, SquisherJoy (see Materials and methods and *Figure 5—figure supplement 1*). For IFT, the outline of the wrapped tissue was used as an approximation for the deep tissue. n's = 4–9 for mesodermal and ectodermal IFT; n's = 4–7 explants for mesodermal, 1–5 for ectodermal TST; error bars = SEM; the same explants were measured at sequential time points, roughly 1 hr apart. Axisymmetric drop shape analysis (ADSA)(*David et al., 2009*; *Luu et al., 2011*) of explants was also used to measure surface tensions. Wontons made with rhodamine dextran labeled deep tissues wrapped in unlabeled epithelium were allowed come to equilibrium and develop for a time (*Video 13*), then fixed, the wonton cut in half and LSCM used to capture the profile of each half (**B**). ADSA of the profile (**E**) was used to determine the IFT between the deep and superficial tissues (**H**); n's = 3–8 explants per time point; error bars = SEM. To measure TST with ADSA, deep cell aggregates were prepared as above (D, left panel), then the profile of the aggregate under 1 g was imaged at sequential time points using a 45° mirror (D, right panel; *Video 14*\*). ADSA was used on outlines of the aggregate to determine the TST of each aggregate over time (**I**); the mean of TSTs for dorsal and ventral IMZ tissue (DIMZ, VIMZ) was plotted over time (**I**); n's = 4–6; error bars = SEM. The MicroSquisher was also used to determine the amount of compression force that wrapped deep tissue (IMZ wontons (Mesoderm) and AC wontons and sandwiches (Ectoderm)) could generate in 30 min (**G**); each point represents a single run.

The online version of this article includes the following source data and figure supplement(s) for figure 5:

**Source data 1.** *Figure 5C, F-I* source data: IFT, TST or force vs. time.

**Figure supplement 1.** Example of parallel plate compression test.

**Figure supplement 2.** Test of epithelial thickness; test of surface tension vs. strain; test of IMZ pushback; behavior of deep IMZ plus NIMZ tissue explants.

**Figure supplement 2—source data 1.** *Figure 5—figure supplement 2C-D* source data: Epithelial thickness vs. time; IMZ, AC surface tension vs. strain.

**Figure supplement 2—source data 2.** *Figure 5—figure supplement 2E* source data: Long push-back force vs. time.

**Figure supplement 3.** Cytoskeleton and Cell motility.

**Figure supplement 3—source data 1.** *Figure 5—figure supplement 3F-J* source data: Cellular movement rate, circularity, length-width ratio, perimeter change rate and spreading rate.

of IMZ wonton explants beginning between Stages 9.5 and 10 (G-1h to G0), nearly doubling by G + 1.5 h, but no increase in AC sandwich or wonton explants (*Figure 5C*).

Increased IFT could result, for example, from an increase in the affinity of the deep mesodermal cells for each other (i.e. an increased TST of this tissue), or of the superficial epithelial cells for each other, or a decrease in affinity between the deep and superficial cells, independent of changes in self-affinity. We tested the first possibility and found that the TST of mesodermal deep tissue aggregates (as in *Figure 5D and E*, left) remained low until after Stage 10, when it increased more than twofold, while the TST of ectodermal deep tissue aggregates remained low (*Figure 5F*). Thus, the simplest explanation consistent with our data is that the self-affinity of the deep IMZ cells, and hence TST, increases, which in the absence of other changes results in an increased IFT and a decreased spreading of the epithelial layer, whereas the low IFT of the deep ectoderm with respect to the epithelium allows epithelial spreading.

We also measured the TST and IFT of IMZ explants using a modified 'Axisymmetric Drop Shape Analysis' (ADSA) (*Figure 5D and E*; *Luu et al., 2011*). The drop-shape of a cell aggregate at equilibrium (the degree to which it is flattened) represents a balance of forces between TST or IFT and gravity (1 g). Because the specific density of *Xenopus laevis* gastrula tissue is known (*David et al., 2009*), surface tension can be calculated by the Young–Laplace equation using the radii of aggregate curvature, which reflect the pressure difference over an interface between two fluids (See Materials and methods and *David et al., 2009*). The higher the surface tension, the 'rounder' a drop-shaped cell aggregate will appear at equilibrium. To use this approach to measure the IFT between deep IMZ and its

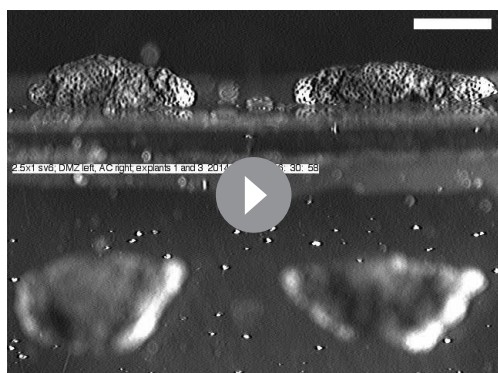

**Video 13.** Wontons rounding up, viewed from the side, parallel to their long axis; out of focus top-down view below. Wontons contain DMZ deep tissue on the left, AC deep tissue on the right. Movie begins at roughly G-1h. 3 min/frame. Elapsed time = 2 hr. Scale bar = 500 μm.

https://elifesciences.org/articles/57642/figures#video13

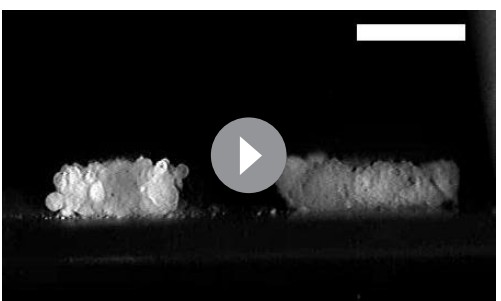

**Video 14.** Aggregates of deep VIMZ (left) and deep animal cap (right) cells rounding up, just after release from cover glass. Movie begins about G + 1 h. 3 min per frame. Elapsed time = 2 hr. Scale bar = 500 μm.
https://elifesciences.org/articles/57642/figures#video14

epithelial layer, we made wontons with deep IMZ tissue aggregates from fluorescently labeled embryos and wrapped them in unlabeled superficial epithelium. Wonton explants were allowed to come to equilibrium for a minimum of two hours at 1 g (e.g. *Video 13*), fixed at specific times, bisected across the longer axis, and images parallel to the cut-face were collected using laser scanning confocal microscopy (LSCM) of the two halves, thereby revealing the 'drop shape' of the deep tissue filling within its epithelial wrapper (*Figure 5B*). As in the parallel plate compression tests, control wonton explants, filled with ecto-dermal (animal cap, AC) deep tissue, did not increase their IFT with respect to the epithelium during gastrulation, whereas wontons filled with deep DIMZ increased their IFT from levels similar to those seen in AC controls prior to the onset of gastrulation, to roughly fourfold higher than the AC controls over the first two hours of gastrulation (*Figure 5H*). We also measured TST from time lapse movies of deep cell aggregates of dorsal and ventral IMZ (*Video 14*) using ADSA; these showed little change in TST until about Stage 10 (G0), at which point both began to increase through Stage 10.5 (G + 2 h) (*Figure 5I*), when the DIMZ tissue plateaued, consistent with the onset of a transition to CE.

Whereas our TST measurements by parallel plate compression were in agreement with previous TST measurements of these tissues (*David et al., 2009*; *Luu et al., 2011*; *David et al., 2014*), our ADSA measurements were about 10-fold lower for both TST and IFT (compare *Figure 5C and F* to H,I). We therefore consider our parallel plate compression TST and IFT measurements to better reflect the state of these tissues in the embryo. A possible explanation for differences in our ADSA measures with prior measures is that our culture media (DFA) was different than used in previous studies (MBS). DFA has a higher specific gravity than MBS, by about 0.7% and has about a 9% higher osmolarity than MBS, which could affect TST measurements of deep cell aggregates (*Krens et al., 2017*), and possibly IFT measurements of wontons, although the epithelium heals around them to form a barrier against diffusion such that their interstitial environment should reflect in vivo conditions, but neither of these differences in media appear to be large enough to account for the observed difference in surface tension measures. Notably, however, both our parallel plate and ADSA measurements showed similar increases in IFT and TST of mesodermal tissues around the onset of gastrulation.

Together, these results indicate an increase in the TST of deep IMZ tissue during CT, consistent with increased cell–cell affinity within the deep mesoderm. Importantly, the increase in IFT between the deep and superficial epithelial tissues in the IMZ was not seen in the IFT between the deep and superficial tissues of the ectoderm (i.e. animal cap). Moreover, the fact that ectodermal epithelium did not spread over deep ectoderm of the NIMZ prior to G-1h, but did by G0 (*Figure 4C*; Shook, unpublished observations), suggests that affinity between these tissues increases between these times. The increase of IFT between deep and epithelial IMZ would be expected to cause an annular (ring-shaped) IMZ to decrease its surface area, that is to converge and thicken.

The rounding and decrease in deep IMZ tissue surface area in explants, and its decreased support for epithelial spreading, predict that the epithelium might also decrease its basal surface area of interaction with the deep region. With constant cell volume, this decrease in the basal surface area of initially rectangular epithelial cells should translate to an increase in cell height, as previously shown when epithelium is allowed to relax (*Luu et al., 2011*), and change the overall shape of cells to that of a cuboidal epithelium (*Figure 5—figure supplement 2A*). To test this, we measured the heights of IMZ epithelial cells in whole embryos as a control, and in sandwich explants between the onset of gastrulation and the onset of CE. Explant epithelial cell heights were initially not significantly different from those in intact embryos (*P* = 0.18, for both dorsal and ventral, n = 3 or 4), confirming that the explant procedure did not alter cell height. However, explant epithelial cells subsequently thinned by 5–10%, rather than thickening, while whole embryo epithelial cells thinned by 25–30%,

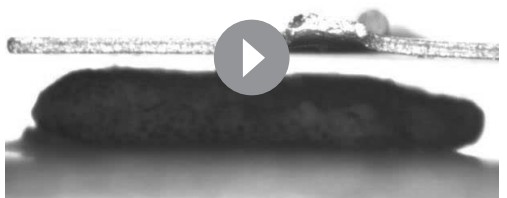

**Video 15.** Example of a long duration pushback experiment. 0.3 µN compression force was applied to a DIMZ wonton (see Materials and methods) over 50 s, after which the plate was left in place to respond passively in order to measure the response of the explant over the next 3 hr. Capture rate = 1 s/frame for first 30 frames, then 100 s/frame for 110 frames, then 1 frame/s for 140 s. Playback = 15 fps. Scale bar = 250 µm.
https://elifesciences.org/articles/57642/figures#video15

as expected (*Figure 5—figure supplement 2B*). These results suggest that the expected epithelial 'slack' in explants is taken up by some other mechanism. Possible explanations are that the apically contracting bottle cells (which we did not measure) are anchored to the vegetal endoderm and stretch the IMZ epithelium vegetally as they contract in vivo (see *Hardin and Keller, 1988*) and/or that epithelial spreading (wound healing) around the uncovered animal and lateral edges may exert tension on the rest of the epithelium, preventing it from thickening in explants.

## CT generates thickening force

The convergence force of CT measured previously (*Shook et al., 2018*) should be accompanied by a thickening force of the same magnitude. Therefore, sandwich explants and wontons of IMZ, NIMZ, and AC were tested in the MicroSquisher for thickening forces (see Materials and methods) immediately after release from the compression applied during healing. We initially applied a light compressive force (0.2–0.3µN) to ensure good contact between the platen and the top of the explant, then measured probe tip displacement without changing the position of the probe base. As the tissue undergoes rounding and thickening (*Video 15*), force is exerted on the platen (*Figure 5G*). The increase in force due to thickening of wontons containing deep IMZ (mesoderm) tested over 30 min, beginning from between G-0.5h to +2.2 h for dorsal IMZ (0.31 µN, SEM = 0.066, n = 7) and from G + 1.3 h to +3.6 h for ventral IMZ (0.27 µN, SEM = 0.023, n = 4), was similar to the previously observed convergence forces from G0 to G + 2 h (about 0.35µN) (*Shook et al., 2018*). AC and NIMZ tissue showed little if any increase in thickening force over intervals beginning from G-1h to G + 3.2 h (0.016 µN, SEM = 0.010, n = 7). Thickening forces were not observed in IMZ explants prior to Stage 10- (G-0.7h; n = 2).

To more precisely determine the onset of thickening force, DIMZ wontons and AC sandwiches were constructed at Stage 8.5–9, such that force testing could begin by Stage 9–9.5 (G-2h to –1 h) and then observed for 3 hr. In most cases, explants showed stress-relaxation as expected of a visco-elastic material, then plateaued for some time before pushing back with increasing force (*Figure 5—figure supplement 2E*). On average, this began at G-41m (G-0.7h, Stage 10-; n = 7, SEM = 1.1 min). In one exceptional case, the explant pushed back almost immediately but then increased its rate of force increase at G-0.7h. The average increase in force from the initial, applied force, or from the force at the onset of increased *rate* of increase at g-0.7h in the exceptional case, was 0.48 µN (n = 5, SEM = 0.11). AC sandwiches instead began to *decrease* the push-back force at G-0.7h (n = 4) (data not shown), perhaps associated with the onset of rapid radial intercalation driving epiboly around this time (*Keller, 1980*), during which deep ectodermal cells actively migrate toward the superficial epithelium (*Szabó et al., 2016*), causing the deep tissue to spread and thin. Note that epiboly as whole begins well before stage 10-; the mechanism driving early epiboly is unknown. The increase in the IFT between deep mesoderm and its overlying epithelium in the IMZ, the corresponding increase in thickening force by the mesoderm, the lack of increase in the IFT between deep ectoderm and its epithelium in the NIMZ and AC regions and the decrease in thickening force by ectoderm at Stage 10- are additional characteristics associated with the onset of CT (see Discussion).

The convergence and thickening force that the measured IFT could theoretically generate is in close agreement with our previously measured convergence forces (here and *Shook et al., 2018*) and the thickening forces measured here. The observed increase in IFT during the onset of gastrulation (G-0.9h to +1.6 h), $\sigma_{em}$ = 0.22 mN/m (*Figure 5C*) acts over the whole interface length $L_i$ in cross-sections of explants (see Materials and methods), that is over the whole contour length of the interface between the deep mesodermal tissue and the surrounding tissue, producing a force

$$F = \sigma_{em} * L_i$$

With a contour length of $L_i$ = 1.2 mm, the expected force of shortening along the long axis of a D180 explant would be $F$ = 0.26 µN, which is indeed close to the *convergence* force measured in the same explants, of about 0.3 µN from G0 to G + 2 h (*Shook et al., 2018*) and our observations here that dorsal IMZ wonton explants can generated about 0.3 µN of *thickening* force over 30 min. Thus, the IFT could be responsible for generating the measured convergence forces during CT.

Our analysis of force generation in terms of surface tension and contour length raises the question of the affinity between deep mesoderm and the adjacent deep ectoderm at the Limit of Involution (LI). In both normal and ventralized whole embryos, this interface was reduced progressively as the circumference of the LI decreased to that of the closed blastopore, whereas in sandwich explants, the interface began to become obviously more constricted by the onset of CE (e.g. Stage 10.5 in *Figure 2A*, *Figure 2—figure supplement 1A*), or by G + 3.5 h in ventralized giants (*Video 3**, ~ 1/3 of the way through). In the case of normal explants, this result can be explained by the CE of both neural and mesodermal tissues. In ventralized explants, these results suggest that the self-affinity of ectodermal and mesodermal tissues exceeds that for each other, with an IFT high enough to drive their separation. However, in explants of deep ventral IMZ +NIMZ (VMZ) under cover glass (Materials and methods), cells from the deep IMZ engulf the deep NIMZ tissue, migrating around the sides of it, and to a lesser extent, over its face, throughout gastrulation (*Figure 5—figure supplement 2F*, VMZ; *Video 16**, *Video 17**). This would be expected if deep mesoderm had a lower TST than deep ectoderm, but our data indicate this is not the case. Apparently, other mechanisms modulate the interaction of these two tissues. One possibility is that at the LI the affinity of the IMZ for the NIMZ is held in check by the attached epithelium.

## CT does not appear to rely on changes in cadherin, actin, or phosphoMLC localization, or in deep cell motility

Given our working hypothesis that CT begins at Stage 10-, we investigated the molecular and cellular basis of the increase in IFT, TST and force production by looking for changes in cadherin, pMLC or actomyosin localization to the superficial-deep interface and in cell motility between Stage 9 and 10. However, immunohistochemistry showed that C-cadherin and actin expression were similar across all types of cell junctions (deep-deep, superficial-superficial, deep-superficial) at both Stage 9–9.5 and Stage 10; pMLC expression was more mosaic, but, on average, also similar across all junctions at both stages (*Figure 5—figure supplement*

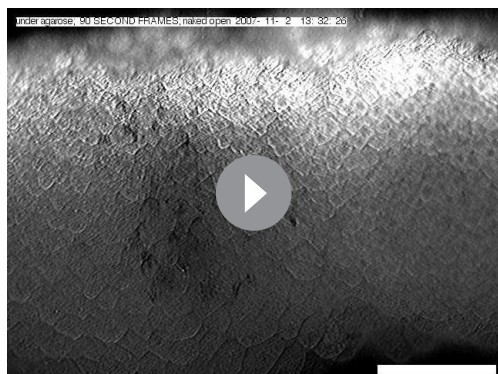

**Video 17.** High-resolution movie of deep ventral NIMZ (top) engulfment by deep ventral IMZ (bottom). Stages 10.25 through 14 (G + 1h to 7.25h). Vegetal toward the bottom. Smaller, more animal IMZ cells show motility over NIMZ region. A small patch of epithelium (cells much less motile) appears on the right side about 1/3 of the way through the movie, moves with the NIMZ and is eventually covered by IMZ cells. Imaged surface is covered by a thin sheet of agarose, under cover glass. Made with a 20 X dipping lens on an Olympus AX70. 1.5 min per frame, 250 frames, 15 fps. Scale bar = 100 µm.
https://elifesciences.org/articles/57642/figures#video17

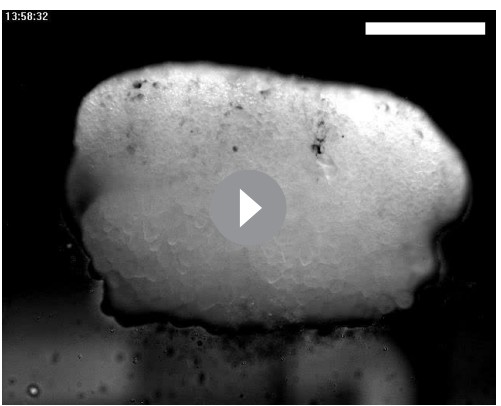

**Video 16.** Naked ventral IMZ showing engulfment. Stages 10.25 through 14 (G + 1h to 7.25h). IMZ cells engulf NIMZ; no reversal at Stage 10.5 (15:00). Under cover glass; imaged at 20 X on an Olympus IX70. 1.5 min per frame, 250 frames, 15 fps. Scale bar = 200 µm.
https://elifesciences.org/articles/57642/figures#video16

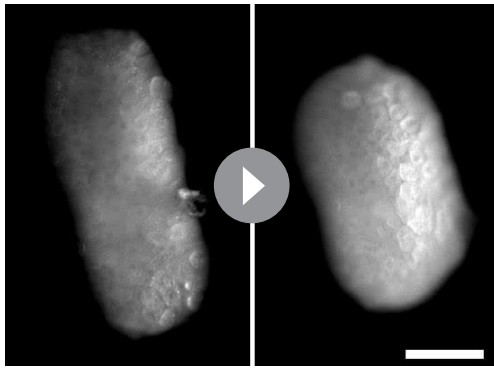

**Video 18.** Explant of deep upper DIMZ, imaged with low-angle illumination. The LR axis is oriented up and down in the movie. Left image is around Stage 9.5 (G-1.5h to G-0.9h), right image is around Stage 10 (G-0.4h to G + 0.2 h). 1 min/frame. Scale Bar = 200 μm.
https://elifesciences.org/articles/57642/figures#video18

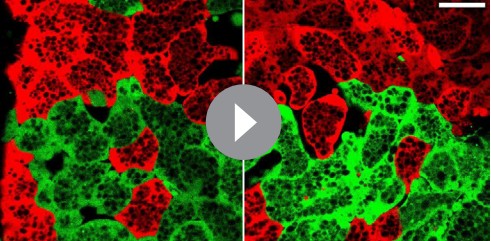

**Video 19.** Explant of deep IMZ from an embryo injected at 2 cells stage with Alexa-555 dextran in one cell and Alexa-488 dextran in the other, imaged at 60 X via LSCM at a depth of 3–5 μm. Movie on the left is around Stage 9.5(G-64m to –49 m), the one on the right around Stage 10 (G-2m to +15 m). 30 s/frame. Scale bar = 20 μm.
https://elifesciences.org/articles/57642/figures#video19

3A-D). These results suggest that some other mechanisms, perhaps affecting the activities or turnover of these molecules, regulate affinity between deep and superficial cells, and therefore the change in IFT between them (see Discussion).

The changing protrusive behavior of mesodermal cells from Stage 10 onward (*Shih and Keller, 1992a*; *Wallingford et al., 2000*), as well as the underlying cytoskeletal dynamics (*Kim and Davidson, 2011*; *Pfister et al., 2016*) have been well characterized. Between Stages 10 and 10.25 (G0 to G + 1 h), cells have rapid, randomly oriented protrusive activity. As cells begin expressing MIB at Stage 10.5, this transitions to mediolateral protrusive activity (*Shih and Keller, 1992a*; *Wallingford et al., 2000*; *Kim and Davidson, 2011*; *Pfister et al., 2016*). Cell motility prior to Stage 10 is less well characterized; beginning at the mid-blastula transition (MBT) at Stage 8.5 (G-3h) partially or completely dissociated cells begin to exhibit "pseudopodal processes" (protrusive activity) and circus movement of blebs, associated with rapid translocation in random directions (*Johnson, 1976*; *Newport and Kirschner, 1982*; *Kimelman et al., 1987*). This onset of cell motility is not dependent on RNA transcription but appears to be the result of the lengthened cell cycle at MBT (*Johnson, 1976*; *Newport and Kirschner, 1982*; *Kimelman et al., 1987*). By Stage 9 (G-2h), 85% of cells exhibit some type of motile behavior (*Newport and Kirschner, 1982*) but it is not clear if this differs from the behaviors characterized at Stage 10 (*Shih and Keller, 1992a*).

To test whether there was any change in cell behaviors associated with the onset of CT around Stage 10- (G-0.7h), we first looked at cell behaviors in deep IMZ tissue explants, either imaged on a compound scope with low angle illumination, as in *Shih and Keller, 1992a*, allowing the observation of 'whole cell' dynamics on the surface of the explant, or labeled with fluorescent dextrans and imaged with LSCM, which gave a higher resolution look at the protrusive behaviors of the cells. Deep IMZ cells show the same type of low-level 'jostling' behavior both before and after Stage 10- (e.g. *Video 18*), with similar levels of compaction at the tissue surface, broken locally as surface cells divided, similar rates of randomly oriented

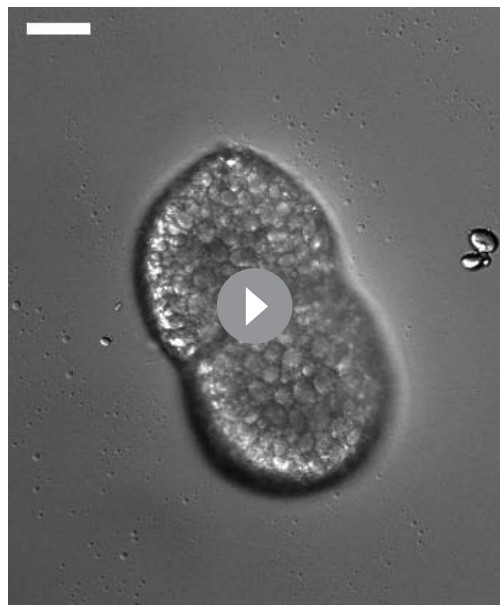

**Video 20.** Dissociated cells transferred onto C-cadherin-Fc substrate, imaged using DIC optics on a Zeiss Axio Observer. G-1.2h to G + 0.8 h. 20–30 s/frame, 6 fps playback. Scale bar = 20 μm.
https://elifesciences.org/articles/57642/figures#video20

cell division and small amounts of random cell movement, with larger movements due to the cell divisions. To observe cell motility at higher resolution, we labeled embryos with green fluorescent dextran on one side and with red on the other, made deep IMZ explants, and imaged them with LSCM around either Stage 9–9.5 or Stage 10 at the interface of the two colors (e.g. *Figure 5—figure supplement 3E*; *Video 19*) to better observe individual cell outlines. Cell outlines were traced every 2–2.5 minutes and several parameters were extracted for comparison. We saw no significant changes for any metric, including movement rate, perimeter change, circularity or length-width ratio (*Figure 5—figure supplement 3F-I*). We also saw no changes in frequency of cell division, net cell movement, orientation of longest axis, or fraction of interstitial spaces (data not shown).

To look for changes in cell dynamics resulting from potential changes in C-cadherin activity around Stage 10-, we plated dissociated cells on the extracellular domain of C-cadherin fused to the Fc domain of IgG (C-cadherin-Fc) at Stage 9–9.5 and imaged them with DIC optics. Whereas mesodermal cells plated on fibronectin will not begin to spread until near Stage 10 (*Ramos et al., 1996*) (Shook, personal observations), cells spread on C-cadherin-Fc as early as tested (Stage 9, G-2h), interrupted by partial contractions and detachment from adjacent cells during cell divisions (*Video 20*). They initially show rapid filiform protrusions as they begin to spread on the substrate, and then a combination of rapid filiform, loboform and lamelliform protrusions as they become more fully spread, or re-spread after cell divisions (*Video 20*); this pattern of protrusive activity did not change over time (data not shown). Spreading continued at a fairly constant rate both before and after Stage 10- (*Figure 5—figure supplement 3J*). No differences in the frequency of cell division were observed before or after Stage 10- (data not shown). These results suggest that changes in cadherin dependent deep cell motility or deep cell contact behavior are not involved in the increase in IFT between deep and superficial cells. Our result that the rate of convergence of the IMZ in unconstrained giant sandwich explants remains constant before and after Stage 10- supports the idea that it is the affinity between superficial and deep, rather than any change in deep cell motility, at least with respect to other deep cells, that is responsible for the increased force production by the IMZ. One possibility is that a change in motility specific to the interface between deep and superficial cells is responsible for this increase (see Discussion), but this interface is technically challenging to image.

## CT movements begin simultaneously and isotropically throughout the IMZ

To determine the onset of convergence movements of the IMZ, and the resulting blastopore closure and involution of the IMZ, we mapped the movements of the IMZ. The pre-involution IMZ is defined by the LI animally and by the vegetal edge of the bottle cell field or the blastopore lip vegetally, depending on stage (*Figure 1A*, *Figure 6—figure supplement 1*). Pre-involution convergence of the IMZ of normal embryos, measured by tracing fiduciary points across the IMZ surface over time (*Figure 6—figure supplement 2*; Materials and methods), occurs at a similar rate in the dorsal, lateral and ventral regions during early gastrulation (*Figure 6A*, left column; 6B; *Video 1*\*). Convergence begins earlier dorsally (*Figure 6B*, blue line), as the apical constriction of the dorsal bottle cells begins at Stage 10- (G-0.7h) and culminates at Stage 10+ (G + 0.5 h) (see *Hardin and Keller, 1988*). The progression of apical constriction of the bottle cells begins about an hour later around the lateral and then ventral sides of the blastopore (*Figure 6A*, left column, Stage 10–10.5; *Video 1*). Although the bottle cells exert tension on the superficial layer above the blastopore, their removal does not change the rate of blastopore closure, which is dependent only on deep IMZ tissue (*Keller, 1981*; *Hardin and Keller, 1988*). The pattern of isotropic convergence of the pre-involution IMZ, excepting the first hour, continues throughout gastrulation, at least until Stage 12.5 (*Keller and Danilchik, 1988*).

There is however a highly anisotropic movement directed toward the blastopore from mid-gastrulation onward (*Keller and Danilchik, 1988*; *Video 1*). This results from mechanical coupling of the pre-involution IMZ to the tissues bordering it, the post-involution mesoderm on the inside and the posterior neural tissues on the outside, both of which are undergoing highly anisotropic CE and thus push the dorsal blastopore lip toward the ventral side of the embryo (see Discussion and *Keller and Danilchik, 1988*; *Figure 6A*, left column, compare Stage 10.5–12; *Video 1*).

In embryos ventralized by UV irradiation of the vegetal hemisphere before first cleavage, Spemann's Organizer is not induced, the dorsal-ventral axis is lost and all tissue types default to ventral (see *Scharf and Gerhart, 1980*). These embryos initiate bottle cell formation at nearly the same time

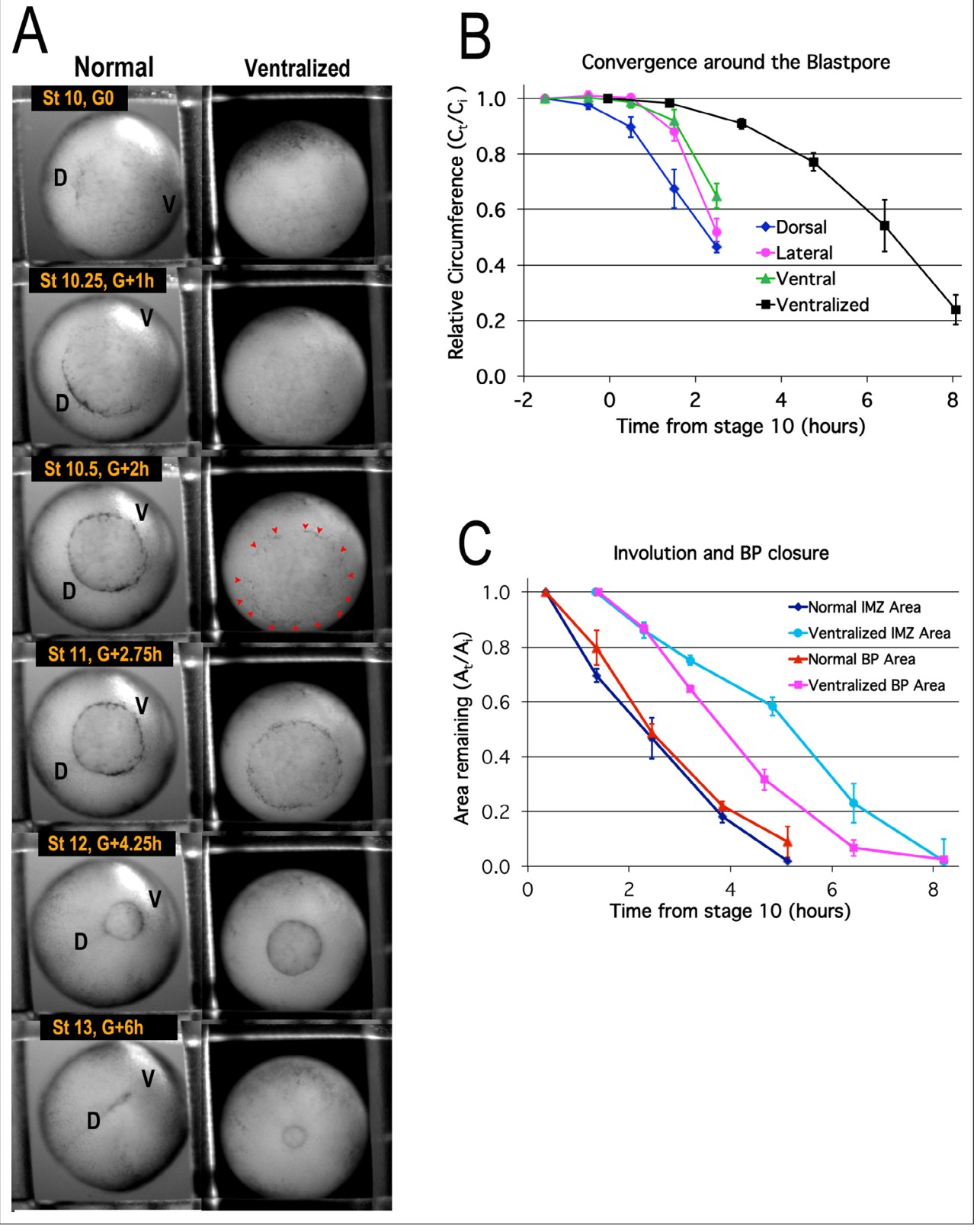

**Figure 6.** Blastopore closure. Time lapse movie frames of normal (left) and ventralized embryos (right) are shown at the stages and times indicated (A, *Video 1\**). Dorsal ('D') and ventral ('V') sides of the embryo are indicated. Blastopore formation (blastopore pigment line formation) and subsequent blastopore closure (see *Figure 6—figure supplement 1* for details) is delayed in the ventralized embryos, with formation of the blastopore usually occurring at control Stage 10.5 (red pointers), two hours after the onset of gastrulation at Stage 10 (G + 2 h). The blastopore of ventralized embryos

*Figure 6 continued on next page*

*Figure 6 continued*

closes symmetrically whereas the normal closure is biased toward the future ventral side (compare Stages 10.5 vs 12). Convergence (**B**) in the dorsal, lateral and ventral quadrants of the pre-involution IMZ of normal embryos (n = 3 or 4 embryos per time point; error bars = SEM within region, across embryos) and in the entire IMZ of ventralized embryos (n = 5 embryos per time point; error bars = SEM across embryos). Change in superficial area compared to its original size ($A_t/A_i$) of the pre-involution IMZ, a measure of its involution, and the area of the exposed vegetal endoderm, a measure of blastopore closure (**C**) (n = 3 normal embryos and 3 ventralized embryos; error bars = SEM across embryos). For both B and C, the same embryos were measured at sequential time points.

The online version of this article includes the following source data and figure supplement(s) for figure 6:

**Source data 1.** *Figure 6B,C* source data: Relative blastopore circumference vs. time, relative blastopore and IMZ area vs. time.

**Figure supplement 1.** The lower (vegetal) and upper (animal) limits of the IMZ are indicated by yellow and green lines superimposed on to vegetal views of a Xenopus gastrula stage embryo at the indicated Nieuwkoop and Faber stages (*Nieuwkoop and Faber, 1967*).

**Figure supplement 2.** Examples of tracking of IMZ and blastopore area through the course of a time lapse movie using fiduciary points.

all around the embryo at about control Stage 10.5 (*Figure 6A*, right column, Stage 10.5, pointers), when normal embryos complete bottle cell formation ventrally (*Figure 6A*, left column, Stage 10.5). Convergence occurs uniformly around the IMZ, driving symmetrical blastopore closure around the VE (*Figure 6*, right column; *Video 1*) but begins later and proceeds more slowly than in normal embryos (*Figure 6B*, black line).

Rates of blastopore closure, based on the area of the exposed VE, are similar in normal and ventralized embryos but show a delay of 1–2 hr in the extent of BP closure in ventralized embryos (*Figure 6C*, cf. red and magenta lines), consistent with the delay of bottle cell formation and convergence in ventralized embryos (*Figure 6A and B*), and suggesting that CE and CT are equally effective at driving blastopore closure. On the other hand, involution of the IMZ in ventralized embryos begins more slowly compared to normal embryos and is delayed by 3–4 hr (*Figure 6C*, light and dark blue lines). This could be due to delayed or reduced Vegetal Rotation (*Winklbauer and Schurfeld, 1999*) in ventralized embryos or indicate that CE is more effective than CT in promoting involution (see Discussion).

These results demonstrate that blastopore closure and convergence of the entire circumference of the IMZ around the blastopore begin well before the onset of CE, that CT accounts for the isotropic convergence of the pre-involution IMZ around the BP, despite the highly anisotropic movements of CE in post-involution mesoderm, and that CT is patterned and can drive BP closure independently of CE.

## CT to CE transition

As the second morphogenic machine operating to close the blastopore, we wished to understand how CE is integrated with CT. Cells begin expressing MIB at Stage 10.5 (G + 2 h), as they progressively involute in the intact embryo, and begin to express MIB at the same time in explants (*Shih and Keller, 1992a*; *Lane and Keller, 1997*). It has been known for some time that the tissues undergoing CE extend vegetally from the thickened band of tissue comprising the IMZ of sandwich explants (see Discussion and *Keller and Danilchik, 1988*), especially in larger, giant explants (*Poznanski et al., 1997*). But the early onset of the thickening and convergence (*Figure 2*), of the changes in affinity between deep and superficial layers (*Figure 4*), and in particular, of the initial force production (*Shook et al., 2018*; *Figure 5G*, *Figure 5—figure supplement 2E*) prior to the onset of CE at Stage 10.5 are definitive evidence that IMZ cells first undergo CT and then transition into expressing CE.

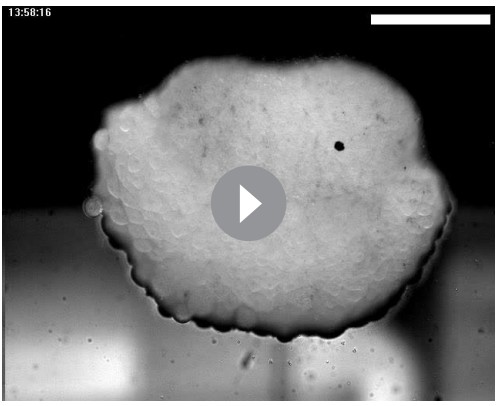

**Video 21.** Naked dorsal IMZ showing engulfment and reversal. Stages 10.25 through 14 (G + 1h to 7.25h). IMZ cells begin to engulf NIMZ, then, at Stage 10.5 (15:00, frame 43), stop and reverse course, eventually converging and extending. Under cover glass; imaged at 20 X on an Olympus IX70. 1.5 min per frame, 250 frames, 15 fps. Scale bar = 200 μm.
https://elifesciences.org/articles/57642/figures#video21

The transition to MIB begins in the presumptive anterior notochordal and somitic tissue, near the dorsal midline (*Figure 1C*, St 10.5) and progresses in an arc-shaped front proceeding animally (the toward the presumptive posterior) in the notochordal mesoderm and laterally (also toward the presumptive posterior) in the somitic mesoderm. It continues to pull cells from the lower edge of the thickened region formed by CT (*Figure 1C*, St 10.5–12+), which remains as a relatively thick 'collar' across the upper, animal portion of the IMZ (that part abutting the vegetal edge of the NIMZ at the LI) throughout gastrulation as the pre-involution portion continues to express CT (*Figure 3—figure supplement 1E*). This progressive transition from CT to MIB results in the progressive anterior to posterior convergence and extension of the axial and paraxial tissue in explants (*Shih and Keller, 1992a*; *Figures 1C and 2A*, *Video 2**), which occurs post-involution in the intact embryo. In the embryo, CT is generating tension around the outside of the blastopore while MIB is generating tension around the inside; these forces act in parallel to close the blastopore and promote involution (See Discussion).

Another behavior associated with the CT to CE transition is seen in deep dorsal IMZ +NIMZ (DMZ) explants lacking epithelium and cultured under a cover glass (see Materials and methods). Just like deep VMZ explants, deep DMZ explants initially show engulfment of NIMZ by IMZ, but in DMZ explants the engulfment is reversed at Stage 10.5, whereas in deep VMZ explants it is not (*Figure 5—figure supplement 2F*; *Video 21**). This indicates a change in cell properties in dorsal but not ventral deep tissue, and since it coincides with the onset of MIB, it is likely to be associated with the transition that cell behavior.

## CT relies on different molecular mechanisms than CE to generate force

To test the idea that CT is dependent on cell behaviors distinct from the MIB driving CE, with distinct molecular controls and effectors, we disrupted two molecules required for MIB, one involved in force generation by MIB and one in patterning the specific cell behaviors of MIB. We expected that normal embryos, which depend on a combination of CT and MIB/CE to generate force for blastopore closure, would be more affected than ventralized embryos, which depend only on CT. An underlying assumption is that MIB is not a process layered on top of CT, but represents a transition *from* the cell behavior driving CT *to* a fundamentally distinct mode of motility resulting from a change in the cells' differentiated state. Thus, if cells still make the transition from CT to MIB/CE, and fundamental parts of MIB are broken, that population of cells will neither continue to generate force by CT nor by MIB (see Discussion).

MIB involves bipolar, mediolateral extension of protrusions, their attachment to neighboring cells via C-cadherin adhesion molecules, and polarized pulsatile actomyosin contractility, in a planar cell polarity (PCP) pathway dependent manner (*Skoglund et al., 2008*; *Kim and Davidson, 2011*; *Pfister et al., 2016*). MIB thereby exerts a cell-on-cell traction that cumulatively generates the mediolateral tension that intercalates the cells and generates tissue level forces (*Skoglund et al., 2008*; *Pfister et al., 2016*). These forces are dependent on MRLC regulation of myosin heavy chain IIB (MHC IIB) contractility (*Skoglund et al., 2008*; *Kim and Davidson, 2011*; *Pfister et al., 2016*). MHC IIB is up-regulated in dorsal tissues expressing CE (*Kelley et al., 1996*; *Bhatia-Dey et al., 1998*), and a morpholino (MO) knock down of MHC IIB strongly inhibits blastopore closure in normal embryos that express CE (*Skoglund et al., 2008*; *Figure 7A*, compare first to second column), whereas the same dose of MHC IIB MO only mildly retards blastopore closure in ventralized embryos (*Figure 7A*, compare third to fourth column)(*Video 22**). MHC IIB MO caused a significant decline in frequency of blastopore closure when injected into normal embryos compared to un-injected controls (*Figure 7B*; $p < 0.001$, n = 61 vs 82); injected normal embryos also had a lower frequency of blastopore closure than injected ventralized embryos (*Figure 7B*; $p < 0.001$, n = 61 vs 31; data from 4 different clutches of eggs). This suggests that CE is strongly dependent on MHC II B, whereas CT is not. Since CT can close the blastopore alone without much MHC IIB, closure is rescued when the transition to CE is blocked via ventralization (*Rolo, 2007*).

Disruption of the non-canonical Wnt/PCP signaling pathway strongly inhibits CE in vertebrates (*Sokol, 1996*; *Heisenberg et al., 2000*; *Tada and Smith, 2000*; *Wallingford et al., 2000*), and in particular, disrupts the polarized MIB (*Wallingford et al., 2000*), CE and blastopore closure in *Xenopus* (*Ewald et al., 2004*). The hypothesis that CT is driven by TST, which need not involve polarized cell behavior, predicts that disrupting PCP should have less effect on tissue movement and

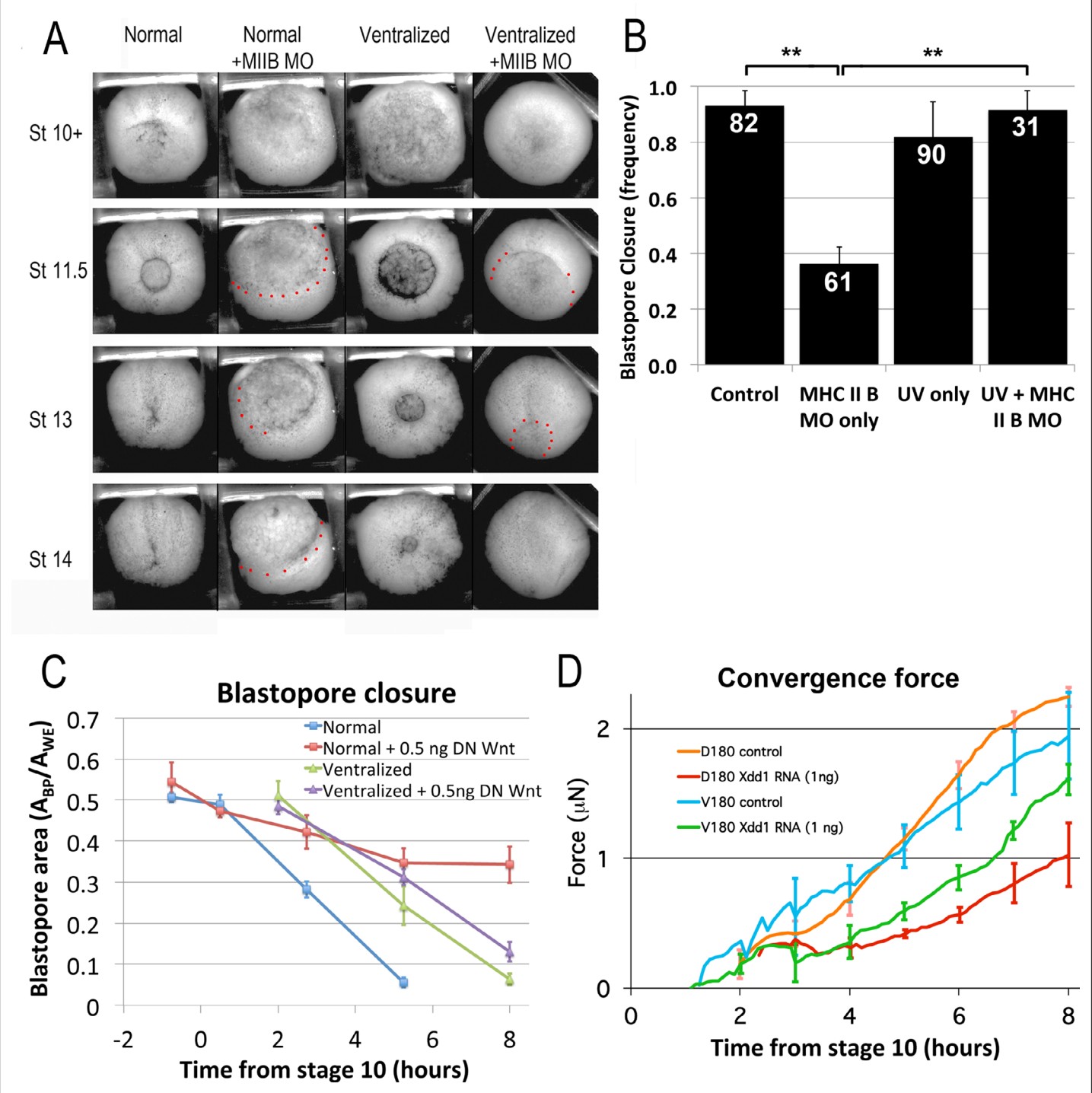

**Figure 7.** CT and CE depend on different molecular pathways. (**A**) Representative stills of time lapse movies (*Video 22*\*) comparing blastopore closure in normal and ventralized embryos, un-injected or injected with an MHC IIB morpholino. The region just outside the blastopore or forming blastopore are indicated by red dots in cases where it is difficult to see. (**B**) Frequency of blastopore closure by Stage 20 (G + 13 h effective success of blastopore closure) was scored for embryos from 4 different clutches of embryos (n's indicated on chart); \*\* = p < 0.01. Comparison of the extent of blastopore closure over time in normal and ventralized embryos, either un-injected or injected with 0.5 ng dnWnt11 RNA (**C**; see for example *Figure 7—figure supplement 1* and *Video 23*\*), based on the projected area of the blastopore, $A_{bp}$, divided by the projected area of the whole embryo, $A_{we}$, measured from time lapse movies (n = 4 embryos for each treatment). Force measurements (*Shook et al., 2018*) of D180 or V180 explants made from embryos, either un-injected or injected with 1 ng Xdd1 RNA (**D**; n's = 3–4 explants per treatment). Errors bars in all cases = SEM.

The online version of this article includes the following source data and figure supplement(s) for figure 7:

*Figure 7 continued on next page*

*Figure 7 continued*

**Source data 1.** *Figure 7B* source data: frequency of successul blatopore closure.

**Source data 2.** *Figure 7C* source data - extent of blastopore closure vs. time.

**Source data 3.** *Figure 7D* source data: Convergent force of D180 and V180 explants.

**Figure supplement 1.** Time lapse movie stills of normal and ventralized embryos, +/- 0.5 ng dnWnt11 mRNA.

force generation driven by CT than by MIB. This idea is supported by the fact that embryos injected with Xdd1 RNA dorsally have greater blastopore closure delays than those injected ventrally (*Ewald et al., 2004*). We found that whereas normal embryos injected with 0.5 ng dnWnt11 RNA (*Tada and Smith, 2000*) had slowed or completely stalled blastopore closure, dnWnt11 RNA injected ventralized embryos showed only a minor delay in BP closure (*Figure 7C*; *Figure 7—figure supplement 1*; *Video 23*\*). Further, Dorsal 180° (D180; e.g. *Videos 24 and 7*) sandwich explants injected with 1 ng Xdd1 RNA (*Sokol, 1996*) showed significantly decreased force generation compared to controls, by 65% at the end of gastrulation at G + 6 h (p < 0.001, n = 3 vs 3; compare red and orange force traces, *Figure 7D*). D180 explants express only CT early but then progressively transition to CE from G + 2 h to about G + 7 or 8 h. In contrast, injection of Xdd1 RNA did not reduce force generated by V180 sandwich explants (e.g. *Video 8*) significantly (41% decrease at G + 6 h; p = 0.08, n = 4 vs 3; *Figure 7D*, compare green and blue force traces); V180 explants only express CT, with no transition to CE. These results show that force production by CT does not depend on PCP signaling as strongly as CE does. That Xdd1 injected D180s generate 60% less force than un-injected V180s supports the conclusion that the D180s do not continue to express CT after going through the transition to MIB, even when MIB motility is blocked. Taken together, these results show that when cells expressing CT subsequently transition to the differentiated state required for CE but are unable to express normal CE motility (MIB), blastopore closure will fail (see *Transition to CE* section in Discussion).

## Discussion

*Scharf and Gerhart, 1980* observed closure of blastopores in ventralized embryos, which do not form the dorsal tissues that undergo CE, implying that some other, previously unidentified mechanism of closure in ventralized embryos. This classic paper was the intellectual and experimental inspiration for the present study. Here we characterize CT as the isotropic, symmetrically acting, force-generating morphogenic machine that can close the blastopore without CE. Our findings here establish CT as a bona fide morphogenic movement, distinct from CE, that functions in blastopore closure. This constitutes a substantial revision of how amphibian gastrulation occurs (*Figures 1 and 8*).

Our key observations are that the 'convergent thickening' initially described as a slight thickening dorsally prior to radial intercalation and a thickening associated with involution ventrally (*Keller and Danilchik, 1988*) is, in fact, a major, independent morphogenic movement: (1) it begins earlier and thickens more than previously observed, and occurs uniformly throughout the entire pre-involution IMZ, throughout gastrulation (See *Onset of CT* section, below), until progressively transitioning to other cell behaviors during involution (See *Transition from CT to CE* section, below); (2) it acts only prior to involution and in parallel with CE, which acts only post-involution (see *Model* section, below); (3) it can act independently of CE to close the blastopore (See *Onset of CT*, *Transition from CT to*

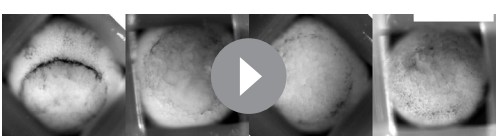

**Video 22.** Blastopore closure in normal and ventralized embryos, either uninjected or injected to 10 μMolar MHC II B MO. From left to right: Normal; Ventralized; MHCIIB MO injected; Ventralized and MHCIIB MO injected. Moive starts at about Stage 10.5 and continues through neurulation. Scale bar = 1 mm.

https://elifesciences.org/articles/57642/figures#video22

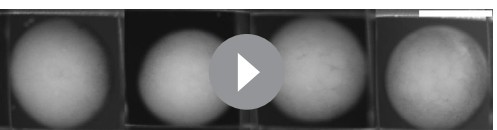

**Video 23.** Blastopore closure in embryos untreated, injected with 0.5 ng dnWnt, ventralized or ventralized +injected with 0.5 ng dnWnt (left to right). Movie starts at Stage 10- (G-0.7h) and continues through late neurulation (~Stage 18). 5 min per frame. Scale bar = 1 mm.

https://elifesciences.org/articles/57642/figures#video23

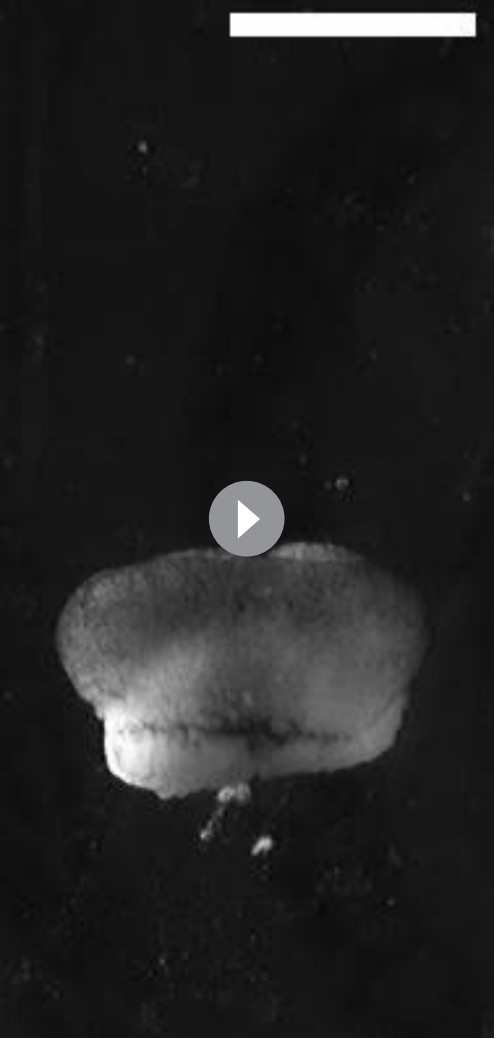

**Video 24.** Dorsal 180° sandwich explant. Movie begins at G + 2.2 h. 3 min per frame. Elapsed time = 13.4 h. Scale bar = 1 mm.

https://elifesciences.org/articles/57642/figures#video24

*CE* and *Evolutionary conservation of CT* sections, below); (4) it is driven by forces resulting from increased interfacial tissue surface tension (see *Tissue surface and interfacial tensions as force generating mechanisms* section, below); (5) it depends on different molecular mechanisms than those required for CE (see *Transition from CT to CE* section, below).

Our characterization of CT also raises issues about the systems mechanics of the multiple morphogenic machines that contribute to blastopore closure. It hopefully will provide a path to resolve how these machines interact, and why perturbation of so many different processes can abrogate blastopore closure (See Model section, below). It also suggests that CT is a conserved feature of anuran gastrulation, and perhaps metazoan morphogenesis in general, and that a comparative approach to understanding how morphogenic machines can be integrated will be useful (see Evolutionary conservation section below).

## Tissue surface and interfacial tensions as force generating mechanisms in morphogenesis

*Holtfreter, 1939* proposed that the movement and arrangement of tissues in the gastrula of amphibians is governed by developmentally regulated, tissue-specific 'selective' affinities, which could be positive or negative. *Townes and Holtfreter, 1955* extended this concept to the behavior of dissociated cells, and Steinberg introduced the Differential Adhesion Hypothesis (DAH) (*Steinberg, 1970*; *Steinberg and Takeichi, 1994*; *Steinberg, 2007*) a thermodynamic model of the process that attributed these behaviors to differential adhesion, emphasizing differences in the strengths of adhesion and the minimization of adhesive-free energy. This model implies that tissues can have surface and interfacial tensions (TST and IFT), in analogy to the respective tensions of fluids. TST is generated by the tendency in a cell aggregate to minimize the free energy by maximizing the adhesive interactions of its cells, thereby minimizing the surface area of the tissue and compacting it into a sphere, and by the tendency among cells with different affinities to form adhesive interactions with similar cells, and so self-segregate (*Holtfreter, 1939*; *Townes and Holtfreter, 1955*; *Steinberg, 1970*; *Steinberg and Poole, 1981*; *Steinberg and Takeichi, 1994*; *Forgacs et al., 1998*). In this sense, tissues mimic the behavior of fluids, with a few important differences (*Harris, 1976*). In particular, tissue fluidity requires active cell motility in order for cells to rearrange themselves (*Armstrong, 1989*). Moreover, TST is also determined by cellular cortical tension, in addition to the kind and number of adhesion proteins cells express (*Brodland, 2002*; *Lecuit and Lenne, 2007*; *Manning et al., 2010*; *Amack and Manning, 2012*; *Maitre et al., 2012*; *David et al., 2014*; *Winklbauer, 2015*), and by Eph/ephrin regulation of cell contact affinities (*Rohani et al., 2011*; *Canty et al., 2017*). These concepts successfully explain the observed segregation of unlike cells, aggregation of like cells, and the rounding up of irregularly shaped cell aggregates in vitro.

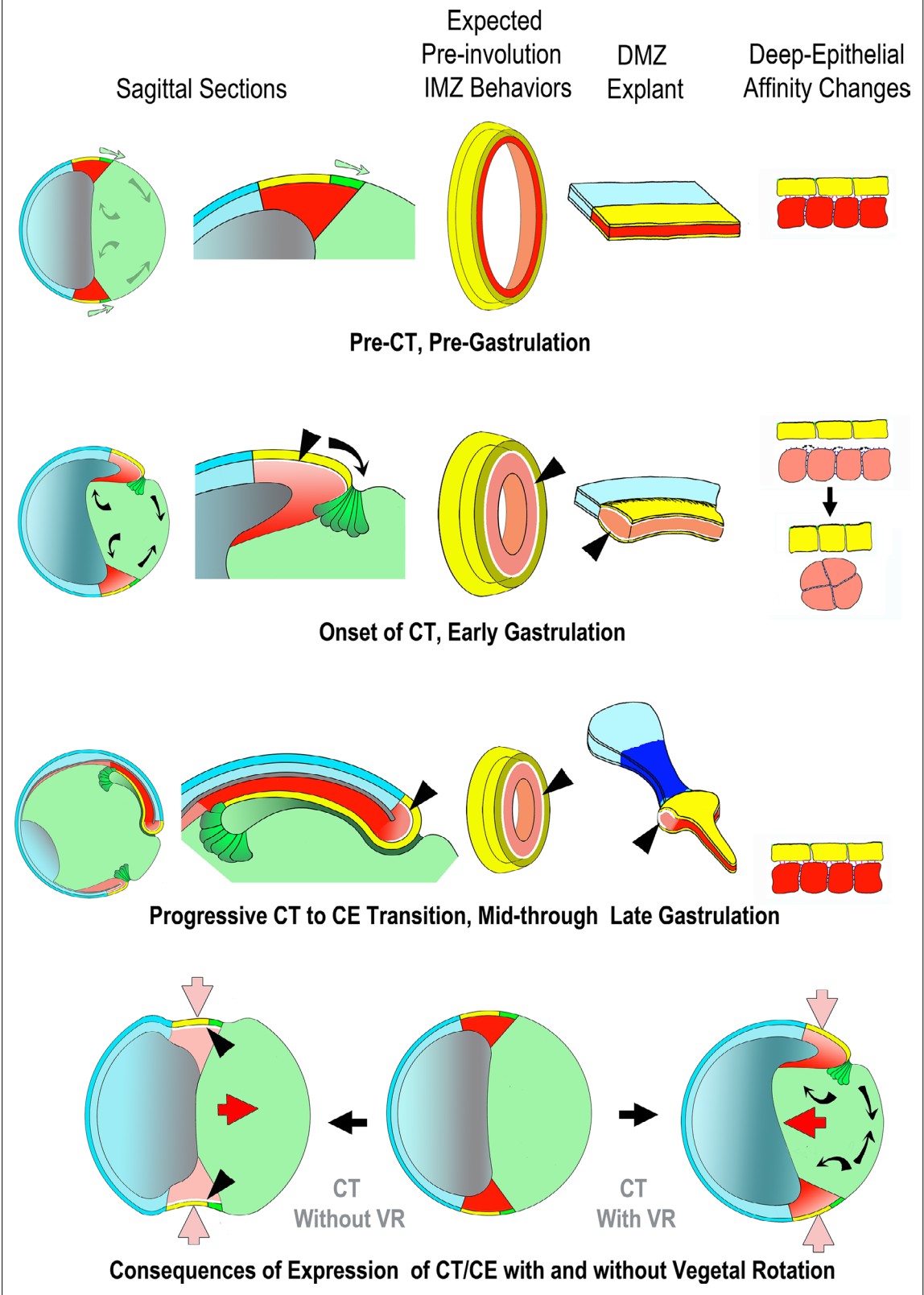

**Figure 8.** Proposed mechanism of CT, its function in the gastrula and in explants, its relationship to CE and its integration with the other movements of gastrulation. Yellow/green = endoderm, pink/red = mesoderm, blue = ectoderm (See *Figure 1* for details). Prior to the onset of gastrulation (top row), vegetal rotation (VR) (Sagittal Sections, gray arrows) has not yet occurred and the IMZ annulus (3rd column) remains stable, as does the IMZ explant (4th column), due to a stable deep-epithelial cell affinity (far right). With the onset of gastrulation and CT (2nd row), apical constriction of the lower IMZ

*Figure 8 continued on next page*

*Figure 8 continued*

epithelium (bottle cell formation – Sagittal Sections, dark green), VR, involution and invagination begin (Sagittal Sections, black arrows). Co-incident with bottle cell formation, the deep mesodermal tissue begins to lose its affinity with the overlying epithelium. During early gastrulation both tissue surface tension of the IMZ as a whole and its interfacial tension with an epithelium increase, resulting in a tendency for the IMZ to minimize its surface area by converging and thickening (2nd row, far right). This loss of affinity at the interface of deep and epithelial tissue (2nd and 3rd rows, shown by white spaces between epithelial and deep, at black arrowhead pointers) increases the IFT between deep and superficial layers, resulting in both convergence and thickening of the IMZ of the explants (CT, second row, bulge in DMZ explant). Increased IFT would have a *tendency* to thicken and decrease the circumference of the IMZ annulus in embryos (3rd column, 2nd row) were it not resisted by adjacent tissue, so instead generates a pre-involution tensile force around the blastopore and converges, but does not thicken (not shown). In explants, the weak affinity of the IMZ epithelium to the underlying deep region results in its retraction, initially from the leading edge mesoderm (not shown), whereas in the intact embryo, the IMZ epithelium is tightly attached at its vegetal edge to the forming bottle cells, which in turn are tightly attached to the adjacent vegetal endodermal tissues. Thus the IMZ epithelium is pulled along with the bottle cells as they move inside the blastopore (1st, 2nd column). During early involution, as the lower edge of the IMZ involutes, it transitions from expressing CT to expressing directed migration (2nd column, 2nd row), and during later involution, the upper IMZ transitions to expressing CE (2nd column, 3rd row). In the case of CE, this transition is accompanied by regaining a high deep-epithelial affinity (loss of white space, on involution, 2nd column, 2nd and 3rd rows). In explants, the later transition from CT to CE first occurs in tissue laying dorsally at the vegetal edge of the upper IMZ, and continues in an anterior to posterior progression (*Shih and Keller, 1992a*). The snout of tissue extending from the thickened collar of mesoderm in the explant reflects the onset of CE and is coincident with the progressive return of the affinity between the deep cells and the bottle cell epithelium (3rd row, 4th column). In the meantime, VR (black arrows, 1st column, 2nd row) has moved the vegetal endoderm inward at the onset of gastrulation such that convergence force generated by CT cooperates with VR to move the IMZ vegally and push the vegetal endoderm inside (compare to 1st column, 1st row). VR, CT and CE also cooperate to stretch the ectodermal (NIMZ-AC) region by pulling it vegally (1st column, 1st to 3rd rows). When CE begins at Stage 10.5, it has been positioned by the actions of CT and VR such that it further moves the IMZ vegally and pushes the vegetal endoderm inside (4th row, right, red arrow), in contrast to converging in its initially more equatorial position, where it would tend to push the vegetal endoderm out, rather than in (4th row, left, red arrow).

The question is whether these concepts can also be meaningfully applied to the morphogenesis of tissues with more complex shapes in the developing embryo. Morphogenic movements are often produced by polarized, anisotropic cellular processes, which notably evolved in the context of tissue affinity and the resultant TSTs and IFTs that may work against them (reviewed in *Winklbauer and Parent, 2017*). In contrast, the significance of our findings lies in the fact that changing the IFT within the context of a particular geometry, an annulus (the IMZ) laying below the equator of a sphere in the intact embryo, results in tissues within the IMZ minimizing the contact area between them, an intrinsically isotropic process that generates oriented, directional forces to a morphogenic movement. In this case, the surface minimization of the deep IMZ generates a circumferential convergence force squeezing the annulus closed, and being below the equator, pushing itself toward the bottom of the sphere, thereby pulling the attached ectodermal tissue down over the sphere (see *Model* section, below).

Our measurements show that the IFT of IMZ tissue increases from the onset of gastrulation while that of NIMZ tissue remains low. The observed decrease in affinity between epithelia and deep IMZ are also consistent with an increase in IFT in the IMZ, but not in the NIMZ, and with the hypothesis that CT is driven by increased IFT in the IMZ, starting at Stage 10-. In the annular IMZ of the whole embryo, increased IMZ IFT would shorten the circumferential dimension of the IMZ, generating a tensile preload force until balanced by the reaction force of adjacent tissues (see *Model* section, below). Forces from other sources, such as adhesion to extracellular matrix or active cell movement may also counteract IFT or modify its effect. The simplest explanation for the increase of IFT consistent with our data is that the deep mesoderm increases its TST, while that of the epithelium's inner surface does not adapt accordingly (see also *Onset of CT* section, below).

Our results support the idea that changes in tissue surface tension can generate enough morphogenic force to drive the movements observed in the embryo. This does not rule out the possibility that some other mechanism may be involved in generating a portion of the observed convergence force during the first two hours of gastrulation. One possibility is that the deep cells are engaged in a different, or additional type of motility, associated with the convergence required for involution (e.g. *Evren et al., 2014*). Additional sources of tension generation could come from the epithelium, for example by its contraction. Although polarized behaviors cannot be eliminated, they would have to be relatively insensitive to PCP perturbation. Additionally, live time-lapse imaging of deep cells prior to the onset of MIB shows no polarized activity (*Shih and Keller, 1992a*; *Pfister et al., 2016*). Our results emphasize the importance of mechanical context in morphogenic function. They suggest that unoriented, unpolarized processes, perhaps arising from changes in general properties such as tissue

affinity and tissue surface or interfacial tensions, expressed in specific geometries and mechanical contexts, may result in directional and polarized morphogenetic forces more commonly than has been appreciated.

## Onset of CT

Tissue movements, biomechanical properties and force generation related to CT that we can measure with good temporal resolution appear to begin at Stage 10- (G-0.7h) and others are consistent with this timing. Prior to Stage 10-, both IMZ and NIMZ regions of sandwich explants converge at about the same rate; thereafter the NIMZ slows its convergence. Prior to Stage 10-, endogenous or transplanted epithelium remain relatively static on both IMZ and NIMZ regions; after Stage 10-, epithelium covering the IMZ begins to contract, while that covering the NIMZ spreads. These changes in the IMZ are correlated with an increase in IFT between its deep and superficial layers at Stage 10-, although the observed increase in TST is somewhat delayed. Force generation by IMZ tissue, as assayed by the ability to exert an increase in thickening force, also consistently begins at Stage 10-, whereas this force begins to decline in NIMZ tissue. Together with earlier observations that dorsal bottle cell formation (see *Hardin and Keller, 1988*), dorsal leading edge mesoderm internalization (*Nieuwkoop and Florshutz, 1950*) and the rapid, active radial intercalation associated with epiboly (*Szabó et al., 2016*) each begin at or around Stage 10-, our results suggest that that a common developmental switch is triggered at Stage 10-. Thus, although many of these changes, clearly expressed in explants, remain largely cryptic in intact embryos, Stage 10- (G-0.7h) marks the true onset of behaviors associated with gastrulation movements. We envision this 'switch' as a single signal or a set of related signals, but these changes could result from unrelated signals with a common timing mechanism.

The onset of CT at Stage 10- appears linked to the activation of the basal mesodermal program by maternal VegT and subsequent Nodal signaling, via phospho-Smad2 (*Kofron et al., 1999*; *Kavka and Green, 2000*; *Lee et al., 2001*; *Xanthos et al., 2002*), since it does not occur in embryos treated with the small molecule inhibitor sb-505124 (Shook, personal observation) and the inhibition of Nodal expression by pharmacological or genetic means blocks both blastopore formation and all subsequent gastrulation movements (*Horb and Thomsen, 1997*; *Kofron et al., 1999*; *Luxardi et al., 2010*). CT is independent of Spemann's Organizer and the downstream dorsal patterning pathways responsible for turning on CE, since it still occurs in ventralized embryos. From its expression in the IMZ and role in mesoderm specification, Brachyury is an obvious candidate for regulation of CT and repressing its transcriptional targets causes an apparent delay in blastopore closure by Stage 10.5, but does not block blastopore closure (*Conlon et al., 1996*), suggesting that other transcription factors are also likely to play a role (e.g. zygotic VegT /antipodean [*Stennard et al., 1996*; *Stennard et al., 1999*;*Kavka and Green, 2000*; *Fukuda et al., 2010*], derriere [*Sun et al., 1999*]; eomesodermin [*Ryan et al., 1996*]). Our results suggest that the most obvious way to screen for candidate genes involved in CT would be to screen them by looking for failed or delayed blastopore closure in ventralized embryos.

CT is a force producing morphogenic movement that alone accounts for the early forces produced by the marginal zone prior to the onset of CE at Stage 10.5, and can act alone, without CE, to produce the forces within the IMZ that close the blastopore in ventralized embryos. Convergence forces measure up to 2 µN in ventralized giant explants, similar to those measured in standard giants during normal gastrulation that result from the combined expression of CT and CE (*Shook et al., 2018*), although the latter may reflect the limitations of the measurement. CT is a pre-involution process, consistent with the largely isotropic convergence movements of the pre-involution IMZ throughout gastrulation in the embryo, as previously described (*Keller and Danilchik, 1988*), and is observed throughout the portion of the IMZ in explants that has not yet transitioned to CE movements, and throughout the pre-involution IMZ of ventralized explants for several hours after the onset of gastrulation (see *Figure 2B*). Whereas CT thickens the IMZ in explants, this does not occur in the intact embryo (See *Model* section).

Our results show that force generation by CT results from increased IFT between the deep and superficial epithelial layers of the IMZ as a result of the decreased affinity between them but the mechanism underlying the decrease in affinity is not yet clear. Loss of affinity in tissue recombination assays between mesodermal deep cells and epithelia occurs regardless of epithelial origin (animal cap, NIMZ or IMZ), suggesting that the change in affinity is a change on the part of the IMZ deep cells with respect to the superficial epithelium. Deep IMZ cells begin to gain a basal level of motility once

the cell cycle is extended at the mid-blastula transition (MBT; Stage 8.5, G-3h) (*Johnson, 1976*; *Satoh et al., 1976*; *Newport and Kirschner, 1982*; *Kimelman et al., 1987*). We were unable to discern any changes between Stages 9 and 10 in cell motility, and prior time-lapse imaging of explants shows only similarly un-polarized cell behavior in the IMZ until the onset of CE and its characteristic, polarized cell behavior, MIB (*Shih and Keller, 1992a*; *Wallingford et al., 2000*; *Kim and Davidson, 2011*; *Pfister et al., 2016*). Together with the equal convergence of the IMZ that we observed in unconstrained giant sandwich explants before and after Stage 10-, these results suggest that a change in deep cell motility with respect to other deep cells does not explain the increase in IFT between superficial and deep.

We also saw no change in C-cadherin, cortical actin or pMLC expression levels at the deep cell-superficial cell interface or elsewhere within the IMZ, suggesting that changes in expression levels of these molecules are likewise not responsible for the changing affinity and that some other mechanism regulating superficial-deep cell affinity must be in play. Cell affinity may be dependent on cadherin activity level at the cell surface, which can be modulated in a number of ways (*Gumbiner, 2005*) including the rate of endocytic turnover (*Ogata et al., 2007*; *Wirtz-Peitz and Zallen, 2009*). Cell affinity can also be controlled by Eph receptor-ephrin ligand signaling, for example during tissue boundary formation (*Fagotto et al., 2014*; *Winklbauer and Parent, 2017*; *Fagotto, 2020*). The decrease of EphB1 and increase of ephrinB2 and B3 around the early gastrula stage (*Rohani et al., 2014*) could be well suited to modulate epithelial-deep cell attachment.

Eph receptor-ephrin ligand signaling generally controls cell interactions at interfaces and could in principle also be responsible for minimizing the interface between IMZ and NIMZ, which dramatically decreases during gastrulation. It is not clear whether increased IFT between deep mesodermal and ectodermal tissues plays a role here, or why un-involuted deep mesoderm migrates over deep ecto-derm in explants (*Videos 16 and 17*) but does not in intact embryos. One possibility is that the deep ectoderm simply out competes the deep mesoderm for attachment to the ectodermal epithelium, as a result of active migration of the deep ectoderm toward the superficial layer associated with epiboly (*Szabó et al., 2016*). Fibrillar FN basal to the deep ectodermal tissue also plays a role in active radial intercalation (*Marsden and DeSimone, 2001*). Deep pre-involution mesodermal tissue does not share this tendency to actively migrate toward the epithelium, and apparently does not interpret cues from basal FN in the same way that ectoderm does (*Kwan and Kirschner, 2003*), as illustrated in our giant sandwich explants by the thickened collar region of the pre-involution IMZ abutted to the much thinner posterior NIMZ.

Our conclusions apply most generally to the upper (animal) end of the IMZ containing the presump-tive chordal and parachordal (paraxial) mesoderm, tissues that undergo a transition to CE after CT. We only poorly understand the role of the more vegetal, leading edge mesoderm (presumptive head, heart, ventrolateral mesoderm) in CT (*Figure 1A*, *Figure 1—figure supplement 1A1,B,C*). The leading edge mesoderm transitions from CT to expressing directed, collective migration after involution (*Winklbauer and Nagel, 1991*; *Davidson et al., 2002*; *Weber et al., 2012*). This transition occurs early dorsally, beginning at Stage 10- with the onset of internal involution of the leading edge mesoderm (*Nieuwkoop and Florshutz, 1950*) and with the onset of bottle cell formation within the epithelium that initially over-lies this mesoderm and the apparent loss of affinity between the two. By Stage 10 the leading edge mesodermal cells have involuted in association with VR movements, coming into apposition with the blastocoel roof by Stage 10+ (*Winklbauer and Schurfeld, 1999*), by which time they are able to begin migrating (*Lee et al., 1984*; *Ramos et al., 1996*; *Davidson et al., 2002*). These movements and the transition occur progressively later laterally and ventrally. Because the leading edge mesoderm is more challenging to work within the context of our experiments, we did not test it as extensively and it was largely removed from explants used for TST and IFT measure-ments. However, the leading edge mesoderm shows a clear early drop in affinity with the bottle cell-forming epithelium, suggesting that it generates force by CT.

## Transition from CT to CE

The timing and geometry of CT force generation and movements *in explants* shows that the cell behavior driving CT in the pre-involution region of the IMZ transitions into post-involution behaviors with the same timing and pattern with which involution occurs *in embryos* (*CT to CE transition* in Results section). CE of the notochordal and somitic mesoderm is a highly anisotropic *post-involution*

process in embryos and occurs with the same distinctive anisotropy directly out of the dorsal-vegetal (presumptive anterior) edge of the collar in explants (*Keller et al., 2000*; *Keller et al., 2008*; *Keller and Sutherland, 2020*).

During involution, the cells previously participating in CT transition into mesendodermal migration, if they involute early, and into MIB (driving CE) if later. Although CT begins uniformly and continues isotropically around the entire *pre-involution* IMZ, the onset of mesendoderm migration and MIB are progressive from origins at the dorsal, vegetal (presumptive anterior) edge of the IMZ toward the ventral, vegetal (presumptive posterior) regions of the IMZ, and from these origins, toward the LI. Thus, the population of cells expressing CT progressively declines throughout gastrulation until involution is complete, around G + 7–8 h (Stage 14).

Note that although we show that CT occurs around their entire circumference of the pre-involution IMZ, the original interpretation of CT (*Keller and Danilchik, 1988*) was that CT was primarily a 'ventral' morphogenic machine, since it occurred in sandwich explants the ventral quadrant whereas CE primarily occurred in explants of the dorsal quadrant (*Figure 1B*). Since then however, it has become clearer that the presumptive somitic tissue, a dorsal tissue expressing CE, wraps all the way around the IMZ of the early gastrula (as shown in *Figure 1A–C*) (see revised fate map', *Keller, 1991*; *Wilson and Keller, 1991*; *Shih and Keller, 1992b*), as do presumptive ventral tissues, such that presumptive dorsal and ventral tissues do not lie exclusively in the so-called 'dorsal' (organizer) and 'ventral' (opposite) sides of the embryo. This point was made most clearly in direct cell tracing and mapping studies by Connie Lane and associates (*Lane and Smith, 1999*; *Lane and Sheets, 2000*; *Constance Lane et al., 2004*; *Lane and Sheets, 2005*; *Lane and Sheets, 2006*). Thus, to be clear, all presumptive dorsal tissues originating in the upper IMZ first express CT, then transition to CE as they involute. However, if the non-organizer, ventral half of the embryo is explanted at the onset of gastrulation, it never receives the signals from the organizer that pattern the presumptive dorsal tissues (*De Robertis and Kuroda, 2004*), explaining the result obtained by *Keller and Danilchik, 1988* and the ensuing misconception.

CT is patterned independently of CE and does not depend on the subsequent expression of CE to close the blastopore, as shown by the fact that ventralized embryos express only CT and successfully close their blastopores, albeit in an abnormal (isotropic) way. That blocking the *execution* of CE by knocking down MHC IIB (*Rolo, 2007*) or interfering with the PCP pathway (our data; *Ewald et al., 2004*) will block blastopore closure in normal embryos, but not in ventralized embryos, emphasizes the independence of CT. Put another way, although blocking the upstream 'dorsalization' signal ultimately leads to failure of *patterning expression* of CE in ventralized embryos, it does not affect the ability of CT to close the blastopore because cells expressing CT never attempt to transition to CE and are left free to close the blastopore by an independent process. However, blocking molecules directly involved in the performance or execution of CE (PCP elements or MHC IIB, for example), but *not* its patterning, will block blastopore closure in normal embryos because cells expressing CT will transition to expressing CE; the cells no longer express CT and although CE is correctly patterned, the cells can not execute the motility required to drive CE.

It is important to understand that cells that undergo the transition from CT to CE can no longer express CT, regardless of whether or not CE is impaired. The transition associated with involution in normal embryos involves many changes in differentiated state, some of which must or are likely to preclude CT. CT requires a loss of affinity between deep and superficial tissues, raising their IFT and promoting a decrease in surface-to-volume ratio area of the deep tissue, but CE requires a decrease in IFT between superficial and deep tissue (*Ninomiya and Winklbauer, 2008*), as CE tends to increase the surface area of the deep tissue. Thus CE and CT differ fundamentally with regard to their dependence on the IFT between deep and superficial layers. Other examples of this transition from one differentiated state to another include the loss of Brachyury expression (*Smith et al., 1991*), which allows cell adhesion to FN (*Kwan and Kirschner, 2003*), also required for CE (*Davidson et al., 2006*), as well as the increased endocytosis of C-cadherin (*Ogata et al., 2007*), possibly facilitating a reduction in cadherin activity and thus deep mesodermal TST; reduced cadherin activity is also suggested to be important for effective CE (*Zhong et al., 1999*). Increased FN adhesion and reduced cadherin activity are both likely to interfere with CT. These changes in Brachyury expression and C-cadherin endocytosis are both independent of the PCP pathway.

On the other hand, PCP disruption may have some effect on CT, although evidently not enough to affect blastopore closure. In addition to its role in regulating MIB, the PCP pathway regulates another important transition in differentiated state, the acquisition of tissue separation behavior (*Wacker et al., 2000*; *Medina et al., 2004*; *Winklbauer and Luu, 2008*; *Luu et al., 2015*). There is also evidence that at least in some tissues, the PCP pathway promotes cell-cell adhesion and increases TST (*Dzamba et al., 2009*; *Luu et al., 2015*).

## Model for CT and its integration with other morphogenic machines in *Xenopus* gastrulation

Here we show how CT functions in gastrulation, and how it is integrated with other active morphogenic machines that facilitate, contribute to, or are essential for gastrulation (refer to *Figure 8* legend for details).

Prior to the onset of CT at Stage 10- (G-0.7h) (*Figure 8*, 1st row), the annulus of deep mesoderm in the IMZ has a stable, strong affinity for its overlying epithelium. At the onset of CT (*Figure 8*, 2nd row), the affinity of the deep tissue for the epithelium decreases, resulting in a zone of high IFT between them. The result is a tendency to constrict and thicken the collar of mesoderm, resulting in a convergence force around the embryo; in an isolated explant, where there is minimal resistance to convergence, it results in rapid CT forming the characteristic thickened 'collar' region, initially throughout the IMZ. The collar does not appear in vivo, however, because the mass of vegetal endoderm resists and slows convergence (*Figure 8*, compare 1st and 2nd columns to 3rd column) to a rate compatible with the transitions to post-involution behaviors that remove cells from the thickening IMZ (*Figure 8*, 1st and 2nd columns) (see model in *Shook et al., 2018*) and with the continuing VR that aids in shrinking the vegetal aspect of the endoderm (*Figure 8*, 1st column, compare 1st and 2nd row). Thus, rather than thickening the IMZ in the embryo, CT instead continuously generates a tensile preloading force, ensuring that the IMZ converges as other tissues move out of the way.

CT works with at least four other morphogenic machines involved in blastopore closure, each of which play a role in setting up a 'stable dynamic' of balanced forces within the spherical geometry of the embryo that results in the IMZ moving down over the bottom the sphere, allowing it to reduce its circumference and converge over the VE while the ectoderm spreads over the vegetal hemisphere behind it. First, VR involves the active, autonomous upwelling of the central VE toward the blastocoel floor and contraction of its exterior surface (*Winklbauer and Schurfeld, 1999*; *Ibrahim and Winklbauer, 2001*; *Wen and Winklbauer, 2017*). This a) reduces the vegetal aspect of the VE, decreasing resistance to the convergence of the IMZ over it, b) draws the IMZ downward and c) positions more of the VE higher within the sphere of the embryo and with respect to the converging ring of the IMZ, such that IMZ convergence tends to push the VE further inward. Second, CE acts in parallel with CT, with CE generating hoop-stress around the inside of the blastopore while CT generates hoop-stress around the outside; the resulting convergence of the IMZ also draws the IMZ lower as it converges over the VE. Third, mesendoderm migration continues the movements begun by VR (*Huang and Winklbauer, 2018*) and leader cells actively pull the leading edge mesoderm and vegetal endoderm up across the roof of the blastocoel, exerting tension on more distal mesodermal and endodermal tissues (*Hara et al., 2013*; *Sonavane et al., 2017*). This helps to move these tissues up into the embryo and away from the blastopore, and may explain why interfering with mesoderm migration can delay blastopore closure (*Klymkowsky et al., 1992*; *Nagel et al., 2004*; *Weber et al., 2012*). Finally, epiboly, a morphogenic movement that involves the thinning and spreading of the ectodermal animal cap region (*Keller, 1980*), also reduces resistance to movement of the IMZ vegetally and thus facilitates its convergence. Failure of epiboly can delay or block blastopore closure (e.g. *Rozario et al., 2009*; *Eagleson et al., 2015*), whereas microsurgical removal of the animal cap region temporarily accelerates it, after which it returns to its normal rate (*Keller and Jansa, 1992*); removal of the cap can also rescue blastopore closure blocked by failure of epiboly (e.g. *Rozario et al., 2009*; *Eagleson et al., 2015*). Epiboly in turn appears to depend, either directly (*Beloussov et al., 1990*; *Chien et al., 2015*) or indirectly (*Dzamba et al., 2009*) on tension applied to the animal cap by the IMZ as it moves vegetally and by VR as it shrinks the vegetal aspect of the VE; this tension on the animal cap facilitates the active radial intercalation of the ectodermal deep cells toward (but not into) the superficial epithelium (*Keller, 1980*; *Marsden and DeSimone, 2001*; *Szabó et al., 2016*).

Because CT is a preloading force exerting continuous compression on the large mass of vegetal endoderm, in the absence of the rotational effect of VR moving the relative position of the IMZ and the endoderm, CT would form a hoop of constriction too far animally and squeeze the vegetal endoderm outward (exogastrulation) rather than pushing it inward (*Figure 8*, 4th row). Likewise, without CT, CE would begin too far animally, with similar results. Failure of CE, as opposed to the mere absence of dorsal tissues, can also result in a delay or failure of blastopore closure, as discussed above (Transition to CE). Thus, the integrated operation of VR, CT, CE and epiboly are required for blastopore closure to proceed normally. VR, CT and CE also all play a role in involution, which in turn is important for internalizing the supra-blastoporal endoderm and the presumptive mesoderm and moving it out of the way of the closing blastopore, but the mechanics involved are less well understood.

## Evolutionary conservation of CT and TST-based morphogenic machines

That CT alone can close the BP in ventralized embryos, coupled with evidence from other amphibians, suggests that the CT/CE partnership seen in *Xenopus* during gastrulation and neurulation may be found throughout the anurans, with CT as a conserved feature of anuran gastrulation and variation in timing of the transition from CT to CE (Shook, personal observations, *del Pino and Elinson, 1983*; *del Pino, 1996*; *Benítez and Del Pino, 2002*; *Keller and Shook, 2004*; *del Pino et al., 2007*; *Elinson and del Pino, 2012*). Because it is a simple mechanism based on physical properties, some variation of an annular, TST-based morphogenic machine may be a conserved mechanism among animals with a blastopore. In the best-studied case among amphibians, gastrulation in *Gastrotheca rhiobambae* involves symmetrical closure of the blastopore without any CE. Indeed, the development of a Spemann Organizer, dorsal tissues of notochordal and somitic mesoderm, and their CE are all delayed until after blastopore closure (*del Pino and Elinson, 1983*; *Elinson and del Pino, 1985*; *del Pino, 1996*). More extensive analysis suggests that the mechanism of this closure is, in fact, CT (see *Keller and Shook, 2004*; *del Pino et al., 2007*). Preliminary work on *Eleutherodactylus coqui*, (Shook – unpublished results), a direct developing anuran, as well as a number of other frogs (*Benítez and Del Pino, 2002*; *Del Pino et al., 2004*; *del Pino et al., 2007*; *Moya et al., 2007*; *Venegas-Ferrín et al., 2010*; *Elinson and del Pino, 2012*) likewise suggest that several species within the superfamily Hyloidea delay the expression of CE until late gastrulation or early neurulation, relying on CT as the primary source of force within the IMZ driving blastopore closure. One consistent feature of species with a greater reliance on CT movements is that they take much longer to gastrulate (see *Keller and Shook, 2004*; *del Pino et al., 2007*); in conjunction with our finding that ventralized embryos show a pronounced delay in involution, this suggests that CE my serve to accelerate gastrulation, by hastening involution. The variable use of CE in combination with CT during gastrulation among anuran species may reflect the great variation in reproductive strategy emblematic of this taxon and the consequent variation in selective pressures on their developmental strategies and egg architecture.

Although urodele (tailed) amphibians show convergence prior to the onset of subduction or convergent extension movements (Shook, unpublished observations and *Shook et al., 2002*) it is not clear whether they have a mechanism comparable to the CT of anurans. Most urodeles have a largely single layered marginal zone, and thus if CT occurs, it would involve a single layer of epithelial cells. A model of how a sheet of epithelial cells could columnarize, and thereby 'converge and thicken' was proposed by *Gustafson and Wolpert, 1967*. In this model, lateral contact between cells would be promoted by increase in lateral affinity of the cells for one another and/or a loss of affinity for the basal substrate, for example fibronectin, and thereby lead to columnarization. Indeed, degradation of the basement membrane is an important step in EMT of primitive streak cells in Chick (*Nakaya and Sheng, 2008*; *Nakaya et al., 2008*; *Nakaya et al., 2013*), and a similar mechanism may operate during ingression through the bilateral primitive streak of Urodeles (*Shook et al., 2002*). Embryonic superficial epithelia in *Xenopus* are generally under strain, as a result of both strain by external forces (*Chien et al., 2015*) and their tendency to spread on the underlying deep cells; releasing this strain allows the epithelial cells to columnarize (*Luu et al., 2011*). A mechanism such as this, whereby the epithelium is released from external strain or it's basal end is released from attachment to the underlying extracellular matrix, could be broadly applicable in driving morphogenesis via the convergence of epithelial sheets, for example by converging the single layered epithelium lying around the blastopore of an invertebrate to help close it, or converging cells within the epiblast of an amniote toward the primitive streak as a prelude to ingression through it. Alternatively, the autonomous tendency of the epithelial cells to

columnarize could be increased. Understanding CT, and how versions of it may be used in other amphibians, and in other metazoans generally, is therefore likely to be important for understanding the evolution of gastrulation.

## Conclusion

That morphogenesis involves movements should have made it obvious from the beginning that evolution would harness physical mechanisms to drive these movements (*Newman and Comper, 1990*; *Thompson and Bonner, 1992*) a principle that inspired us to understand the mechanics of morphogenesis. TST is a mechanism that must be at work in all metazoan embryos, since their defining feature is cell-cell affinity, although where it acts as morphogenic machine and where it is a feature that must be overcome by some other morphogenic mechanism is another question, one that we look forward to learning the answer to.

# Materials and methods

### Key resources table

| Reagent type (species) or resource | Designation | Source or reference | Identifiers | Additional information |
|---|---|---|---|---|
| Antibody | pMLC 2 (Ser19) (Rabbit monoclonal) | Cell Signaling | Cat. #3,671 RRID:AB_330248 | (1:100) |
| Antibody | 6B6 (recognizes C-cadherin) (mouse monoclonal) | Developmental Studies Hybridoma Bank | AB_528113 RRID:AB_528113 | (1:300) |
| Other | phalloidin-Alexa 555 | (Acti-stain, Cytoskelton.com) | 8,953 | Derivatized fluorescent dextran (1:100 at 4 C, 12–36 hours) |
| Sequence-based reagent | MHC IIB morpholino | *Skoglund et al., 2008* | | Morpholino 5'CTTCCTGCCCTGGTCTCTGTGACAT3' |
| Sequence-based reagent | Xdd1 | *Sokol, 1996* | | RNA |
| Sequence-based reagent | dnWnt11 | *Tada and Smith, 2000* | | RNA |
| Software, algorithm | NIH Image 1.6 | Wayne Rasband, National Institutes of Health; available at http://rsb.info.nih.gov/nih-image/ | RRID:SCR_003073 | |
| Software, algorithm | Object Image | Norbert Vischer, University of Amsterdam; available at https://sils.fnwi.uva.nl/bcb/Object-Image/object-image.html | RRID:SCR_015720 | |
| Software, algorithm | Image J | http://rsb.info.nih.gov/ij/ | RRID:SCR_003070 | |
| Software, algorithm | Metamorph | https://www.moleculardevices.com/products/cellular-imaging-systems/acquisition-and-analysis-software/metamorph-microscopy#gref | RRID:SCR_002368 | |
| Software, algorithm | SquisherJoy | (CellScale, Waterloo, Canada) | Software for MicroSquisher; RRID:SCR_022034 | |
| Software, algorithm | Axisymmetric Drop Shape Analysis | *del Río and Neumann, 1997* | ADSA in Matlab RRID:SCR_001622 for Matlab | |
| Software, algorithm | curvature outline detection | https://www.mathworks.com/products/matlab.html | (Canny) in MATLAB RRID:SCR_001622 for Matlab | |
| Software, algorithm | Surface tension determination in SquisherJoy | Based on *Brodland et al., 2009*; *Mgharbel et al., 2009* | | |

## Embryo manipulations and microinjection

*X. laevis* embryos were obtained and cultured by standard methods (*Kay and Peng, 1991*), staged according to *Nieuwkoop and Faber, 1967*, and cultured in 1/3 X MBS (Modified Barth's Saline) at 22°C–24°C.

To ventralize embryos, de-jellied embryos were placed in dishes made of 15 mm transverse sections of 60 mm diameter PVC pipe with Saran wrap stretched across the bottom, irradiated 6 or 7 min from below at about 35 min post fertilization on a UV trans-illuminator (analytical setting, Fotodyne Inc Model 3–3500) and left undisturbed for at least an hour to avoid accidentally rotating and thus dorsalizing them (*Black and Gerhart, 1986*). Embryos forming bottle cells asymmetrically or earlier than the majority of ventralized embryos were discarded as being insufficiently ventralized. To dorsoanteriorize embryos, embryos were exposed briefly to Lithium chloride (0.35 M in 1/3 X MBS at the 32 cell Stage, for 6 min)(*Kao et al., 1986*). Control, ventralized or dorsoanteriorized embryos were cultured to control Stage 35–38 and scored for their DAI (*Kao and Elinson, 1988*) to evaluate the effectiveness of ventralization or dorsalization.

To label with rhodamine dextran amine (RDA), embryos were injected at the 1–2 cells Stage with 25–50 ng RDA (Molecular Probes, D1817). Embryos and explants made from RDA labeled embryos were fixed at the desired stages with MEMFA (*Kay and Peng, 1991*). To knock down MHCIIB, embryos were injected with 7.5 pmol of a MHC IIB morpholino (*Skoglund et al., 2008*) per embryo. To disrupt PCP signaling, embryos were injected with 1 ng Xdd1 RNA (*Sokol, 1996*) or 0.5 ng dnWnt11 (*Tada and Smith, 2000*). Xdd1 RNA efficacy in the embryos used for the tractor pull was confirmed by an absence of CE in dorsal tissue, either in the sandwich explant in the tractor pull, or tissue removed from the explant, for D180 and V180 explants, respectively.

## Explant preparations

Explants are made and cultured in Danilchik's for Amy (DFA) (*Sater et al., 1993*), which is designed to mimic the osmolarity, ionic content and pH of interstitial fluid, minimizing artifactual effects on surface tension, as have been seen elsewhere (*Krens et al., 2017*). For explants made before Stage 10 the embryos were tipped and marked to identify the dorsal side (*Sive et al., 2000*). A graphical illustration of sandwich explant construction and composition is provided (*Figure 1—figure supplement 1*), with details in the legend. Sandwich explants are so-called because two of the regions shown, which include both deep and superficial epithelial layers, are sandwiched deep side to deep side, such that the superficial epithelium heals across the cut edges, making an epithelial seal around the entire explant, thereby avoiding healing artifacts in subsequent assays. Explants were in general held in place by placing a fragment of cover glass cut to size, with silicon grease on either end, over the explant and pressing gently such that the explant was stained to roughly +10–20% of its apparent area. In most cases, the cover glass was removed after healing.

Naked dorsal or ventral IMZ + NIMZ explants were made by peeling the superficial epithelium off the dorsal or ventral-most ~90° of the embryo at Stage 9–10, cutting the peeled sector out of the embryo, trimming away vegetal endoderm and some leading edge mesoderm from the vegetal edge of the IMZ, and pressing the remaining tissue under cover glass, allowing observation of the lateral engulfment behavior. In some cases, the explant was covered with a thin sheet of agarose in between the surface being imaged and the glass, to reducing sticking and observe more native cell motility.

For epithelial spreading assays, the inner face of the explant was briefly pressed against FN-coated glass to which it rapidly adhered, and a strip of superficial epithelium was laid along the animal to vegetal extent of the explant. The epithelium was held in place with cover glass to prevent it from curling up on itself and to promote adhesion with the deep tissue; after about 30 min, the glass was removed. Attachment to FN prevented explant rolling, which otherwise frequently obscured the epithelial patch. In other cases, the deep tissue was placed on a BSA-coated substrate and a patch of epithelium was laid across the NIMZ/IMZ boundary and held in place with cover glass.

To make deep tissue aggregates, superficial epithelium was removed from AC, dorsal or ventral IMZ regions at Stage 8.5–9.5, and the exposed deep tissue cut out and trimmed down to make sure it included no other tissue type. Clumps of deep tissue were then placed in agarose wells to promote aggregation and rounding for 30–60 min (*Figure 5D*). To make wontons, such aggregates were wrapped in excess superficial epithelium and pressed under cover glass, such that the edges of the epithelium healed together all around the deep tissue (*Figure 5A*). Animal cap sandwiches were made by combining rectangles of ectoderm from the animal-most 30° of the embryo. For surface tensions measures (below), the glass was removed after healing and wontons or sandwich explants were allowed to equilibrate on a flat surface for at least 1 hr.

During explant construction involving the IMZ, explants converge rapidly to about 75% of their mediolateral (circumferential) extent in the intact embryo over the 2.5–3 min it takes to cut out and make the explant (*Shook et al., 2018*), as expected from prior work showing that the IMZ is under strain in the embryo (*Beloussov et al., 1990*; *Chien et al., 2015*). Further convergence slows or halts when the two halves of the sandwich are pressed together under a fragment of cover glass to allow healing. We report relative thickness from the time the cover glass is lifted, normalized to explant width just after construction, prior to being pressed under the cover glass. Thus, our measures of convergence do not capture the initial ~25% convergence with respect to their initial mediolateral extent in the embryo.

Comparing the average IMZ thickness in histological sections of embryos (doubled in *Figure 3— figure supplement 2B*, to be comparable to sandwich explants) to rough measurements of IMZ explants just after explanation, explants thicken during explantation by about 50%. They return to roughly their initial thickness in the embryo when the sandwich is pressed together under a cover glass. During the course of lifting the cover glass, cutting out the boundary regions and healing together the newly cut edges, the explants re-thickened by about 50%. Thickness is reported from the time the two IMZ or NIMZ regions heal together and can be imaged from the side, and thus does not capture the initial ~50% thickening of the IMZ with respect to its initial thickness in the embryo.

## Imaging

Explants were imaged through stereomicroscopes with CCD cameras, using low-angle epi-illumination; for side views, a surface mirrored 45° prism was used. Images were captured using NIH Image. Bottom views of embryos or higher resolution views of cellular behaviors in explants were imaged on an Olympus IX70 or AX70 with image capture via Metamorph (https://www.moleculardevices.com; RRID:SCR_002368). RDA labeled embryos and explants were generally bisected, cleared in Murray clear (2:1 benzyl benzoate: benzyl alcohol) and imaged parallel to their cut face using LSCM. Some sandwich explants or deep cell aggregates made from embryos labeled with fluorescent dextrans were imaged en-face by LSCSM.

## Immunohistochemistry

Actin was visualized in embryos fixed in 3.7% formaldehyde +0.1% glutaraldehyde in TBS +0.1% for 10 min and stained with phalloidin-Alexa 555 (Acti-stain, Cytoskelton.com) at 1:100 at 4 °C overnight to 2 days. Cadherin and pMLC were visualized in embryos fixed with MEMFA (*Kay and Peng, 1991*), and stained with 6B6 (Developmental Studies Hybridoma Bank, AB_528113; RRID:AB_528113) at 1:300 and pMLC 2 (Ser19) antibody 3,671 (Cell Signaling; RRID:AB_330248) at 1:100 at 4 °C overnight to 2 days.

Yolk autofluoresence was used in some cases to visualize cells, so that gaps between them could be more easily visualized.

In all cases, vitelline envelopes were removed prior to fixation and embryos were bisected along the animal-vegetal axis with a scalpel after rinsing out of fix, dehydrated gradually in isopropanol for actin staining or methanol for Cadherin/pMLC and imaged in Murray clear for LSCM.

## Morphometrics and image processing

Explant morphometrics were done with NIH Image 1.6 software (Wayne Rasband, National Institutes of Health; available at http://rsb.info.nih.gov/nih-image/; RRID:SCR_003073), Object Image (Norbert Vischer, University of Amsterdam; available at https://sils.fnwi.uva.nl/bcb/Object-Image/object-image.html; RRID:SCR_015720) or Image J (http://rsb.info.nih.gov/ij/; RRID:SCR_003070).

To measure deformation of tissues over time, high contrast points (e.g. a darkly pigmented cell) on the surface of embryos or explants were tracked through time lapse sequences using Object Image or ImageJ and proportional deformation was calculated as a fraction of the initial size. For measures of marginal zone and vegetal endoderm areas on embryos, apparent (2D projected) areas were corrected for the spherical geometry of embryos.

To measure the interface length (Li) between mesoderm and non-mesodermal tissues in sandwich explants, RDA labeled sandwich explants cut across their long (mediolateral) axis were imaged parallel to their cut faces with LSCM and the images used to trace the boundary between the deep

mesodermal tissue and the overlying superficial epithelium or the adjacent deep ectodermal or endodermal tissue.

## Surface tension measurements

Tissue surface tension was quantified using the Axisymmetric Drop Shape Analysis (ADSA in MatLab; RRID:SCR_001622) algorithm (*del Río and Neumann, 1997*) modified for use with tissues by introducing more robust image parameters to accommodate the irregular curvatures of aggregates (*David et al., 2009*; *Luu et al., 2011*; *David et al., 2014*). For RDA-labeled wontons, two to six confocal slices at 10 µm intervals were projected using max intensity to generate an aggregate outline of the labeled deep tissue.

The curvature outline was identified using an edge detection algorithm (Canny) in MATLAB. All processed images were inspected prior to measurements to ensure that Canny detections generated accurate depictions of the tissue aggregate. ADSA is a program written for MATLAB that numerically integrates the Laplace equation to generate theoretical drop-shapes for hypothetical surface tensions, which are optimally fitted onto experimental profiles obtained from tissue aggregates. The drop-shape that best fits the aggregate is used to derive the specific tissue surface tension for that particular aggregate. Explants that were too irregular for ADSA to determine a drop shape were not used.

The Young–Laplace equation of capillarity defines the equilibrium of a liquid surface as the following:

$$\Delta P + \Delta \rho g z = \gamma \left( \frac{1}{R1} + \frac{1}{R2} \right);$$

from left to right, $\Delta P$ is the difference in pressure across the interface between the tissue and the immersion medium, $\Delta \rho$ is the difference in density between the tissue aggregate and the immersion medium, $g$ represents gravitational acceleration, $z$ is the vertical distance from the aggregate apex, $\gamma$ is the tissue surface tension, and $R1$, $R2$ are the radii of the aggregate profile. To measure TST using Drop Shape Analysis, tissue aggregates (see above) were transferred onto flat substrates with non-adhesive coating and allowed to equilibrate for 2 hr, then imaged for 4 hr, using a 45° mirror to capture their profile and observe changes in TST over time. To measure IFT between deep tissue and an epithelial layer using Drop Shape Analysis, wontons (see above) made from RDA labeled deep tissue, wrapped in unlabeled epithelium, were allowed to age to specific control stages, then fixed, cut through the middle, cleared, and imaged parallel to the cut surface via LSCM microscopy.

A parallel plate compression tensiometer, the MicroSquisher (CellScale, Waterloo, Canada), was also used to measure surface tension. This follows the same general principles as described above, but with a force applied between two plates, rather than by gravity. The ellipsoid shape of the tissue after 3 min of stress relaxation, as determined using the MicroSquisher-associated software, SquisherJoy (RRID:SCR_022034), was then used to determine the surface tension (*Brodland et al., 2009*; *Mgharbel et al., 2009*) as follows:

$$\sigma = \frac{F}{\pi R_3^2 \left( \frac{1}{R_1} + \frac{1}{R_2} \right) - 2\pi R_3 \sin(A)}$$

where $\sigma$ is the surface tension, F is the applied force at 3 min, $R_1$ and $R_2$ are as above, $R_3$ is the radius of the area in contact with the plates and $A$ is the contact angle of the aggregate with the plate (For example, see *Figure 5—figure supplement 2*). Aggregates of deep tissue or wontons, prepared as above but without RDA label, and smaller (60–120°) sections of IMZ, NIMZ or animal cap sandwich explants were used to assay TST with respect to the media or IFT with respect to epithelium. Wontons and sandwich explants were oriented such that the profile obtained was a cross-section of their long axis. Explants for which profiles were not adequately clear or which had highly irregular profiles were not used. Our measures of wontons in the MicroSquisher differ from the drop shape analysis method in that the profile of the epithelium encasing the deep tissue was used, rather than the profile of the deep tissue itself.

## Tractor pull force measurements

Forces generated along the mediolateral axis by sandwich explants were assayed using the tractor pull apparatus (*Shook et al., 2018*). More recent experiments were done using 250 µm glass beads, as they generate less friction with the sled than 100 µm beads.

## Statistical analysis

Excel (Microsoft) was used to organize data and compute basic statistics. For comparison between treatments, a t-test was used, two-tailed, un-paired. Error bars in all graphs are standard error of the mean (SEM). For all plotted average measures, the arithmetic mean was used.

## Acknowledgements

This work was inspired by Professor John Gerhart, University of California, Berkeley, in the early 1980s, who posed the question: "If convergence and extension closes the blastopore, how do you suppose the ventralized embryo, which does not do convergence and extension, closes its blastopore?". NICHD R37 HD025594 MERIT Award to Ray Keller, NIH RO1 GM099108 to Paul Skoglund, NIH RO1 GM094793 and R35 GM131865 to Doug DeSimone, CIHR MOP-53075 to Rudi Winklbauer. We acknowledge the Keck Center for Cellular Imaging for the use of the Zeiss 510 microscopy system (PI:AP; NIH-RR021202)

## Additional information

### Funding

| Funder | Grant reference number | Author |
|---|---|---|
| Eunice Kennedy Shriver National Institute of Child Health and Human Development | NICHD R37 HD025594 MERIT award | Ray E Keller |
| National Institute of General Medical Sciences | NIH RO1 GM099108 | Paul Skoglund |
| National Institute of General Medical Sciences | NIH RO1 GM094793 | Douglas W DeSimone |
| National Institute of General Medical Sciences | R35 GM131865 | Douglas W DeSimone |
| Canadian Institutes of Health Research | CIHR MOP-53075 | Rudolf Winklbauer |

The funders had no role in study design, data collection and interpretation, or the decision to submit the work for publication.

### Author contributions

David R Shook, Conceptualization, Data curation, Formal analysis, Investigation, Methodology, Project administration, Resources, Supervision, Validation, Visualization, Writing – original draft, Writing – review and editing; Jason WH Wen, Conceptualization, Formal analysis, Funding acquisition, Investigation, Methodology, Supervision, Writing – review and editing; Ana Rolo, Formal analysis, Investigation, Methodology, Visualization; Michael O'Hanlon, Conceptualization, Funding acquisition, Investigation, Methodology, Project administration, Resources, Supervision, Visualization, Writing – review and editing; Brian Francica, Investigation, Visualization; Destiny Dobbins, Douglas W DeSimone, Funding acquisition, Investigation, Resources, Writing – review and editing; Paul Skoglund, Funding acquisition; Rudolf Winklbauer, Conceptualization, Funding acquisition, Investigation, Methodology, Supervision, Writing – review and editing; Ray E Keller, Conceptualization, Funding acquisition, Methodology, Project administration, Resources, Supervision, Writing – review and editing

### Author ORCIDs

David R Shook http://orcid.org/0000-0002-0131-1834
Jason WH Wen http://orcid.org/0000-0001-7402-5073
Douglas W DeSimone http://orcid.org/0000-0003-1926-1588
Rudolf Winklbauer http://orcid.org/0000-0002-0628-0897
Ray E Keller http://orcid.org/0000-0001-5686-1959

## Ethics

This study was performed in strict accordance with the recommendations in the 8th Edition of the Guide for the Care and Use of Laboratory Animals, of the National Institutes of Health. All of the animals were manipulated according to an approved institutional animal care and use committee (IACUC) protocols of the University of Virginia. The protocols were approved by the Animal Care and Use Committee of the University of Virginia (protocols #2581 and #1830). All surgery was performed under Tricaine anesthesia, and every effort was made to minimize suffering. The animal care and use program is accredited by the Association for Assessment and Accreditation of Laboratory Animal Care, International. The University of Virginia has a PHS Assurance on file with the Office of Laboratory Animal Welfare (OLAW) (PHS Assurance #A3245-01). The University of Virginia is a USDA registered research facility(USDA Registration # 52-R-0011).

## Decision letter and Author response

Decision letter https://doi.org/10.7554/eLife.57642.sa1
Author response https://doi.org/10.7554/eLife.57642.sa2

# Additional files

## Supplementary files

• Transparent reporting form

• Source data 1. Convergence rates of NIMZ and IMZ in explants.

## Data availability

No large-scale data set were generated. Data upon which figures are based is included as source data for those figures; specifically, there are files for each of Figure 2C-E; Figure 3C,D; Figure 3-figure supplement 1C,D; Figure 3-figure supplement 2B; Figure 4C; Figure 5C,F; Figure 5-figure supplement 2B-D; Figure 5-figure supplement 2E; Figure 5-figure supplement 3F-J; Figure 6B,C; Figure 7B; Figure 7C; Figure 7D. Additionally, there is also a source data file with the data supporting a statement within the results section.

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
