## [Editor Report]

In this manuscript Shook and collaborators investigate the process of convergent thickening, a morphogenetic process occuring within the mesoderm prior to involution during amphibian gastrulation. They demonstrate that convergent thickening occurs in the involuting marginal zone, and acts as an isotropic, symmetrically acting morphogenic process that can close the blastopore. They demonstrate that convergent thickening proceeds before and is independent of convergent extension and is also independent of dorsoventral patterning. Whereas the cellular and molecular basis of convergent thickening remains to be discovered, this process is driven by increased interfacial tension between the deep and superficial epithelial layers of the involuting marginal zone.

---

## [Decision Letter]

**Decision letter after peer review:**

Thank you for submitting your article "Characterization of convergent thickening, a major convergence force producing morphogenic movement in amphibians" for consideration by *eLife*. Your article has been reviewed by 3 peer reviewers, including Lilianna Solnica-Krezel as Reviewing Editor and Reviewer #1, and the evaluation has been overseen by Marianne Bronner as the Senior Editor. The following individual involved in review of your submission has agreed to reveal their identity: Ed Munro (Reviewer #2).

The reviewers have discussed the reviews with one another and the Reviewing Editor has drafted this decision to help you prepare a revised submission.

Summary:

In this manuscript Shook and collaborators extend their recent work published in *eLife* demonstrating that blastopore closure in the frog *Xenopus laevis* depends on the forces generated by cicrumblastoporal convergence of the marginal zone. The current manuscript addresses the tissue mechanics and cellular mechanisms underlying the convergent thickening (CT) of the involuting marginal zone (IMZ), a process that precedes mesoderm internalization. The work combines direct observations and physical measurements/manipulations on explanted tissues to characterize more fully the spatiotemporal expression and cellular basis of CT. The main findings are that CT is an isotropic convergence process, expressed uniformly around the entire pre-involution marginal zone. Moreover, CT shows little dependence on planar cell polarity and Myosin heavy chain IIB, and that CT is driven by an increase in "tissue surface tension" within the deep IMZ cells which, by driving thickening and shortening of a continuous circumblastopore IMZ ring, could explain the tensile convergence force.

There was consensus among the three reviewers that the core message of the paper is important and suitable for *eLife*. The experiments have the unique Keller quality, with clear input from other experts in the field of morphogenesis, and are comprehensive in nature. Generally, the data provide strong support for the author's conclusions. However, the reviewers thought that some conclusions will require additional experimental support. In addition the reviewers thought that the presentation of the data as well as their discussion are not facilitating understanding of the presented results or conveying well the concepts. Therefore, we would like to invite a resubmission of your manuscript, when the following concerns are addressed.

Essential revisions:

1. As currently written, the paper very hard to read. The observations and data that most strongly support the paper's central conclusions are embedded within a larger body of additional observations that bear on how CT is coordinated with other morphogenetic behaviors (e.g. MIB) within the larger process of tissue involution and blastopore closure. The collective significance of these observations can only be appreciated by presenting them in the context of an even larger body of previous work, which the authors have attempted to do. It would be particularly beneficial to restructure the main text so that it is clear how the current study builds on the previous published work (Shook et al., 2018). In addition, the authors need to streamline the presentation of the results to first emphasize those that support a view of CT as a force-producing morphogenetic engine, and that elucidate the core underlying cellular mechanisms (increased TST in the deep IMZ cells). Additional results concerning the relative timing, transitions between, and coordination of CT and MIB, plus differences in the underlying molecular regulation, could be separately streamlined and presented as a body at the end of the paper.

2. It would be very helpful if the authors could provide a more thorough introduction to the terms TST and IFT, and the relationship between them. Throughout this part of the manuscript, the authors move back and forth between TST and IFT, often in the same sentence, in a way that is inherently confusing.

3. It was difficult to see "The retraction of the epithelium from the underlying deep IMZ" in Figure 4C or 5A. This is an important observation in support of the thesis that there is a decreased affinity between deep and superficial IMZ tissues during CT". This needs to be documented in a more compelling way, even if "affinity" is used in the broader, Holtfreterian sense.

4. The authors focus on the role of contact with the superficial epithelium in determining the overall CT force produced by the deep IMZ tissue, but clearly, attachment to the NIMZ will also limit the force produced by deep IMZ cells. Couldn't this also be measured in a parallel plate assay?

5. Whereas the experiments presented in Figure 6A, B can be interpreted that the epithelium has lower affinity to the anterior and posterior IMZ compared to NIMZ, those presented in Figure C are less convincing: the amount of epithelium below the green line is small to start with, and therefore its proposed retraction is not compelling. And how the green line was drawn in the course of the time lapse?

6. The measurements of TST and IFT using ADSA and parallel plate compression yield results that differ by more than an order of magnitude, even though the qualitative trends over time or between tissue types are the same in both assays. The authors present both results in Figure 6, and make no attempts to explain this discrepancy, although they acknowledge that parallel plate measurements agree much better with their previous measurements (Shook et al., 2018) of tensile convergent forces produced by CT. Is there a way to resolve this discrepancy? If not, does it make sense to lead with a set of results that the authors believe least?

7. The resistance of the superficial epithelium to deformation of the deep IMZ will depend both on the deformability of the epithelium, and the strength of attachment between them, which limits shear across this interface. Do compression measurements sample the relative contributions of attachment and deformability differently in wonton explants vs sandwich explants?

8. It would be interesting to see what drives the increase of tissue surface tension at the cellular and subcellular levels. For example, quantitative characterization of cellular dynamics on explants during convergent thickening would help understanding the increase of the tissue surface tension increase between stages 10 and 10.5.

9. Similarly, it would be interesting to see the changes at the level of cytoskeletal proteins during convergent thickening and the possible impact of cytoskeleton perturbation on the process. Currently, the authors provide characterization of the convergent thickening at the tissue level, yet it is clear that convergent thickening depends on the cellular behaviors. Partially, authors addressed the importance of cellular morphodynamics by perturbing Myosin IIB and planar cell polarity regulators but again there is no clear explanation of what is happening at the cellular level when it comes to these perturbations.

10. Thinking about potential molecular mechanisms that could underlie the increased TST of the IMZ, consistent with increased cell-cell affinity within deep mesoderm, are there any candidate emerging from numerous gene expression profiling, including scRNA-seq, experiments carried out during frog gastrulation? This should be at least discussed.

---

## [Author Response]

Essential revisions:1. As currently written, the paper very hard to read. The observations and data that most strongly support the paper's central conclusions are embedded within a larger body of additional observations that bear on how CT is coordinated with other morphogenetic behaviors (e.g. MIB) within the larger process of tissue involution and blastopore closure. The collective significance of these observations can only be appreciated by presenting them in the context of an even larger body of previous work, which the authors have attempted to do. It would be particularly beneficial to restructure the main text so that it is clear how the current study builds on the previous published work (Shook et al., 2018). In addition, the authors need to streamline the presentation of the results to first emphasize those that support a view of CT as a force-producing morphogenetic engine, and that elucidate the core underlying cellular mechanisms (increased TST in the deep IMZ cells). Additional results concerning the relative timing, transitions between, and coordination of CT and MIB, plus differences in the underlying molecular regulation, could be separately streamlined and presented as a body at the end of the paper.

We have re-written the introduction such that the contributions of the prior work, and in particular Shook et al., 2018, that form the basis of our investigation should now be clear – see paragraph starting near the bottom of page 4 (line 20). In particular, Shook et al., 2018 showed 1) ventral or ventralized IMZ tissue generates convergence force and 2) the entire IMZ generates convergence force prior to the onset of CE, which together show that CT is a morphogenic machine independent of CE. Further, Shook et al., 2018 also suggested that CT occurred throughout the pre-involution IMZ but did not demonstrate this, and made the argument that CT transitions to CE at involution, all points that we confirm here.

The results have also been reorganized to present concisely our data showing that CT is associated with a loss of superficial – deep affinity (Figures3 and 4), leading to an increase in IFT, which leads to a thickening force (Figure 5), both corresponding to the onset of CT (points made throughout, summarized in Discussion).

2. It would be very helpful if the authors could provide a more thorough introduction to the terms TST and IFT, and the relationship between them. Throughout this part of the manuscript, the authors move back and forth between TST and IFT, often in the same sentence, in a way that is inherently confusing.

TST and IFT have been more clearly defined at the start of the Results section titled “Interfacial Tension and Tissue Surface Tension in the IMZ increase during CT.” A more thorough explanation can also be found in the Discussion section titled “Tissue surface and interfacial tensions as force generating mechanisms in morphogenesis.” We have also checked to make sure we are using the two terms appropriately throughout.

3. It was difficult to see "The retraction of the epithelium from the underlying deep IMZ" in Figure 4C or 5A. This is an important observation in support of the thesis that there is a decreased affinity between deep and superficial IMZ tissues during CT". This needs to be documented in a more compelling way, even if "affinity" is used in the broader, Holtfreterian sense.

The retraction of the epithelium seen in Figure 4C of the previous submission (now Figure 3A; former Figure 5A has been dropped) can be seen far more clearly in the video from which the stills in the figure are taken (Video 9). Our videos are not mere “supplementary data” but in many cases the most compelling illustration of our results, a point we now make at the beginning of the Results section. Arrow heads have been added to current Figure 3A to indicate the edge of the epithelium before and after retraction. The wholesale and abrupt wrinkling up of the superficial tissue and the reduction in area of contact between deep and superficial tissues is pretty strong evidence that the deep IMZ cells are not a good substrate for epithelial spreading.

Additionally, we consider the contraction of the exogenous epithelia over the IMZ shown in current Figure 4B and the retraction of the exogenous epithelium away from the IMZ shown in Figure 4E, in contrast to the spreading of the epithelia over the NIMZ in either case, to be the most compelling evidence for the loss of affinity, a point that is readily apparent in the videos that the figures are based on (Videos 11, 12).

4. The authors focus on the role of contact with the superficial epithelium in determining the overall CT force produced by the deep IMZ tissue, but clearly, attachment to the NIMZ will also limit the force produced by deep IMZ cells. Couldn't this also be measured in a parallel plate assay?

The edge-wise interface between the deep IMZ and the NIMZ in the sandwich explants is a mystery. The dorsal NIMZ is presumptive neural tissue, with a very, very wide and short presumptive spinal cord next to the IMZ, and that region undergoes a very rapid and vigorous CE, so great that the number of cells intercalating in from far lateral regions into the midline push many of the original midline inhabitants far forward into the hindbrain to the point that original midline cells may be rare in the midline of the extended neural axis As neural CE narrows and extends the NIMZ away from the IMZ, the contact between them is greatly reduced, as neural CE occurs much faster than the CT mediated convergence of the posterior IMZ bounding it. During early neurulation, as the last of the thickened CT region transitions to CE of the posterior notochordal and somitic tissue, rapid intercalation also occurs at the mesodermal (IMZ) side of the interface with the neural tissue, further reducing their contact in sandwich explants.

In the intact embryo, however, the deep IMZ and deep NIMZ cells would be converging and extending in parallel, with the latter above the former, and they are kept separate by Tissue Separation Behavior, a complex process involving Eph-Ephrin regulation of contact behavior, worked out by Rudi Winklbauer and associates. In the sandwich explant, which lack involution, the deep cells of NIMZ and IMZ have edgewise, “on end” contact at the tissue level, and stay separated, but if an explant is bent in any direction at that juncture, the lateral surfaces of these two tissues, bent toward one another, “discover” one another, and rapidly anneal within the epithelium to form what approximates their intended, normal relationship! It is beyond our knowledge to speculate on the dynamic properties of these tissues and how attachment of the IMZ to NIMZ would limit or enhance force produced by the IMZ. It is also not clear how we could measure this in a parallel plate compression test. It is the increase in interfacial tension between deep and superficial IMZ that allows for the generation of tensile forces by the deep tissue.

The situation at the interface between deep IMZ and NIMZ in the intact embryo (at the “Limit of Involution”) is somewhat complicated. This interface is minimized during gastrulation to the circumference of the closed blastopore at end of gastrulation, driven at least in part by the CE of both posterior mesodermal and neural tissue adjacent to the interface, as noted above for explants. Explants of dorsal deep IMZ and NIMZ begin to separate at Stage 10.5, but explants of ventral deep do not (Figure 5-supplementary figure 2F). However, the interface between deep IMZ and NIMZ is also minimized during gastrulation in ventralized embryos, suggesting the hypothesis that the affinity between deep NIMZ and the epithelial layer outcompetes the affinity between deep IMZ and deep NIMZ. These points are explored in the Discussion.

5. Whereas the experiments presented in Figure 6A, B can be interpreted that the epithelium has lower affinity to the anterior and posterior IMZ compared to NIMZ, those presented in Figure C are less convincing: the amount of epithelium below the green line is small to start with, and therefore its proposed retraction is not compelling. And how the green line was drawn in the course of the time lapse?

Again, the video that the figure (now Figure 4E) is based on (Video 12) is more compelling. The green line represents the upper limits of the deep mesodermal tissue, which is migrating toward the top of the explant as the epithelium retracts. The line is based on differential cell size and motility, which are evident in the video. The initial position of the green line within the field of view in subsequent frames is indicated by the open orange line. Comparing the G+1h to the G+6h stills, you can see that the epithelium has actually moved significantly with respect to the initial boundary, which it initially overlapped. These points are now made within the text and figure legend.

6. The measurements of TST and IFT using ADSA and parallel plate compression yield results that differ by more than an order of magnitude, even though the qualitative trends over time or between tissue types are the same in both assays. The authors present both results in Figure 6, and make no attempts to explain this discrepancy, although they acknowledge that parallel plate measurements agree much better with their previous measurements (Shook et al., 2018) of tensile convergent forces produced by CT. Is there a way to resolve this discrepancy? If not, does it make sense to lead with a set of results that the authors believe least?

Differences in the media used could explain part of the discrepancy, but not all of it; this is now noted within the results. We now present the parallel plate measurements first, but still include the ADSA results because they do show the same trend. Another potential explanation is that the explants are under greater strain in the parallel plate compression test than in the ADSA test, and while this shouldn’t matter for a non-living, homogenous substance, it might matter for a living tissue. However, we consider this possibility too speculative to discuss in the manuscript.

7. The resistance of the superficial epithelium to deformation of the deep IMZ will depend both on the deformability of the epithelium, and the strength of attachment between them, which limits shear across this interface. Do compression measurements sample the relative contributions of attachment and deformability differently in wonton explants vs sandwich explants?

Sandwich explants have only the endogenous epithelium, initially leaving the edges exposed. In the case of animal cap/ ectoderm deep tissue, the epithelium will begin to heal together starting at Stage 10-, whereas in the case of deep mesoderm, the epithelium will retract. Thus, mesodermal sandwiches are unusable whereas in ectodermal sandwiches, the epithelium undergoes a small amount of strain, which might affect results. However, we saw no differences between ectodermal sandwiches and wontons (which are much more challenging to make, especially at early stages).

Wonton’s have excess epithelium, which we expected would minimize any resistance to deformation, as any strain could be relieved by recruiting more epithelium from the excess. Conceivably, the lower affinity of mesoderm vs. ectoderm to epithelium could influence the ease of recruitment. However, the epithelium normally undergoes roughly 100% strain during the course of gastrulation, so may not offer much resistance to deformation in any case.

8. It would be interesting to see what drives the increase of tissue surface tension at the cellular and subcellular levels. For example, quantitative characterization of cellular dynamics on explants during convergent thickening would help understanding the increase of the tissue surface tension increase between stages 10 and 10.5.

We asked the reviewers for clarification on this question via the editors, but received none. We looked at cell motility of deep cells at three scales but could see no changes in metrics of cell motility at the level of whole cell displacement, protrusive activity or cell division around the time when other changes associated with the onset of CT occur. This probably means that the basis of CT has to do with changes in the relationship between deep and superficial cells, but we have not yet worked out methods that allow us to look at cell motility at this interface.

9. Similarly, it would be interesting to see the changes at the level of cytoskeletal proteins during convergent thickening and the possible impact of cytoskeleton perturbation on the process. Currently, the authors provide characterization of the convergent thickening at the tissue level, yet it is clear that convergent thickening depends on the cellular behaviors. Partially, authors addressed the importance of cellular morphodynamics by perturbing Myosin IIB and planar cell polarity regulators but again there is no clear explanation of what is happening at the cellular level when it comes to these perturbations.

We looked for changes in actin, cadherin and phospho-myosin light chain localization at the interface between deep and superficial cells, or in comparison to interfaces between deep cells or between superficial cells, but again saw no changes around the onset of CT. Nor do we see differences by cell shape analysis that would indicate a change in the cortical tensions at the interface between deep and superficial. Simply perturbing actin or myosin would decrease or block cell motility and would certainly affect CT, but is unlikely to tell us anything specific about the regulation of interfacial tension.

More sophisticated approaches may reveal the regulatory mechanisms driving increased interfacial tension but are beyond the scope of this revised manuscript. They will probably require differentially treating deep and superficial populations and then recombining them in explants. Preliminary experiments using such an approach show that they are technically challenging, as many such treatments render the epithelium very difficult to work with.

10. Thinking about potential molecular mechanisms that could underlie the increased TST of the IMZ, consistent with increased cell-cell affinity within deep mesoderm, are there any candidate emerging from numerous gene expression profiling, including scRNA-seq, experiments carried out during frog gastrulation? This should be at least discussed.

There are no examples of general gene expression profiling experiments that look specifically at deep or superficial tissues and of the few characterized examples of genes that are expressed specifically in the deep or superficial layer (e.g. Xnr3), they are not expressed in the right pattern at the right time. We have discussed potential mechanisms, namely ephrin/Eph-receptor interactions and other mechanisms of modulating cadherin activity levels without changing cadherin localization.